# Dynamic Reward Incentives for Emergent Cooperation under Changing Rewards

## Abstract

*Peer incentivization* (PI) is a popular multi-agent reinforcement learning approach where all agents can reward or penalize each other to achieve cooperation in social dilemmas. Despite their potential for scalable cooperation, current PI methods heavily depend on fixed incentive values that need to be appropriately chosen with respect to the environmental rewards and thus are highly sensitive to their changes. Therefore, they fail to maintain cooperation under *changing rewards* in the environment, e.g., caused by modified specifications, varying supply and demand, or sensory flaws — even when the conditions for mutual cooperation remain the same. In this paper, we propose *Dynamic Reward Incentives for Variable Exchange* (DRIVE), an adaptive PI approach to cooperation in social dilemmas with changing rewards. DRIVE agents reciprocally exchange reward differences to incentivize mutual cooperation in a completely decentralized way. We show how DRIVE achieves mutual cooperation in the general Prisoner's Dilemma and empirically evaluate DRIVE in more complex sequential social dilemmas with changing rewards, demonstrating its ability to achieve and maintain cooperation, in contrast to current state-of-the-art PI methods.

## 1 Introduction

Many AI scenarios, such as autonomous driving (Shalev-Shwartz et al., 2016), smart grids (Dimeas & Hatziargyriou, 2010), and IoT applications (Deng et al., 2020), can be modeled as self-interested and online learning *multi-agent systems (MAS)*, where conflicts arise due to opposing goals or shared resources (Buşoniu et al., 2010). To maximize *social welfare* (undiscounted sum of rewards across all agents in the system), cooperation is essential in self-interested MAS, requiring agents to avoid selfish behavior for the collective good. This tension between individual and collective rationality is typically modeled as a *social dilemma (SD)* (Rapoport, 1974; Axelrod, 1984). Designing distributed mechanisms to incentivize cooperation in SDs remains a key challenge because it directly affects global efficiency (e.g., aggregated social welfare) and sustainability (e.g., long-run resource availability and fairness across agents) (Axelrod, 1984; Trivers, 1971; Dafoe et al., 2020; Perolat et al., 2017b). Such decentralized coordination is particularly important in ad hoc MAS, where independent agents sharing infrastructure may originate from different manufacturers or stakeholders and thus cannot realistically rely on a global reward signal imposed by a centralized authority. *Multi-agent reinforcement learning (MARL)* is a widely used framework to train rational agents in SDs and temporally extended *sequential SDs (SSD)*, where each agent maximizes its own reward (Leibo et al., 2017; Perolat et al., 2017b; Foerster et al., 2018; Yang et al., 2020).

However, non-cooperative game theory and empirical studies show that naive MARL approaches often fail to sustain cooperation because independent adaptation leads to mutual defection (Axelrod, 1984; Van Lange et al., 2013). Furthermore, policy optimization can be affected by inconsistencies in the reward function, which we summarize as *changing rewards*. Such variations are common when moving from simulation to reality, where physical signals replace abstract reward proxies. Noisy sensors, hardware degradation, shifting specifications, or changing market demands can likewise alter reward scales during training (Dulac-Arnold et al., 2019). In human-in-the-loop settings, evaluators may also adapt feedback online, leading to abrupt or irregular shifts (Knox & Stone, 2009). In these cases, the underlying SSD structure (i.e., greed and fear inequalities) is preserved, but the numerical values fluctuate. Methods that are brittle to such changes struggle to sustain cooperation beyond narrow, hand-tuned conditions.

*Peer incentivization* (PI) enables agents to reward or penalize each other to foster cooperation in SDs (Lupu & Precup, 2020; Yang et al., 2020). It has gained traction for its success in complex SSDs and links to biology, economics, and social science (Vinitsky et al., 2023; Schmid et al., 2021; Kölle et al., 2023). Yet, existing PI methods rely on fixed or domain-tuned incentive values and fail under changing rewards, even if cooperation conditions remain unchanged (Eysenbach & Levine, 2021; Zhang et al., 2020). As Fig. 1c illustrates, prior PI methods might succeed in a given instance of the Prisoner's Dilemma (Foerster et al., 2018; Phan et al., 2024; Yang et al., 2020), but fail when payoffs shift or scale, despite satisfying the same *greed* and *fear* inequalities (Axelrod, 1984; Macy & Flache, 2002). Notably, popular approaches such as *Learning to Incentivize Other learning agents* (LIO) (Yang et al., 2020) or *inequity aversion* (IA) (Hughes et al., 2018) appear superficially capable of adapting to such variation, since they rely on learned or relative incentives. Yet, IA requires precise hyperparameter tuning to be effective and fails to promote cooperation in both the original (unscaled) payoff matrix of Fig. 1c (left) and the uniformly scaled variant (right). LIO achieves cooperation under the original payoffs but degrades under simple scaling, as its learned incentive magnitudes are tied to the absolute reward scale. As a result, these methods either lose robustness under reward shifts or suppress valuable heterogeneity in agent performance. Addressing such shifts typically requires hyperparameter re-tuning, which is impractical in ad hoc or online learning scenarios.

To this end, we propose *Dynamic Reward Incentives for Variable Exchange (DRIVE)*, an adaptive PI framework for SDs with changing rewards. By adapting incentives directly to *reward differences*, DRIVE sustains cooperation without retuning and without conflating legitimate variance with exploitative behavior. We summarize our contributions as follows:

- We introduce a reciprocal exchange mechanism that incentivizes cooperation through reward differences.

- We prove that DRIVE aligns incentives toward mutual cooperation in generalized Prisoner's Dilemma games, remains invariant to reward shifts and scaling.

- We empirically demonstrate DRIVE's robustness and superior cooperation in SSDs under changing rewards.

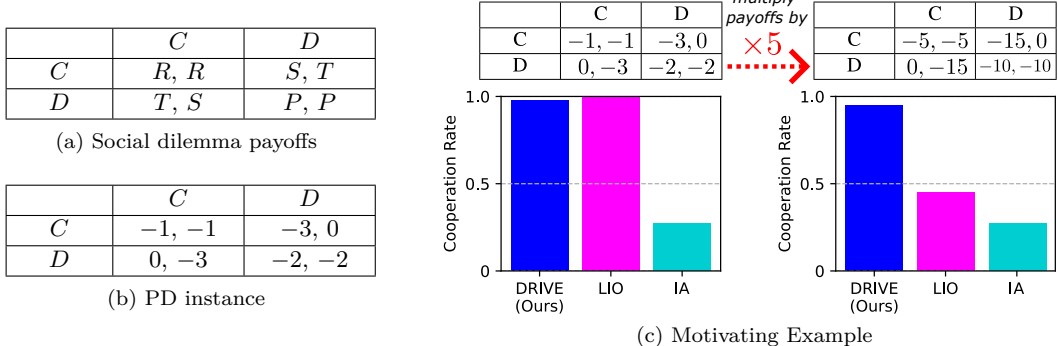

|   | $C$ | $D$ |
|---|-----|-----|
| $C$ | $R, R$ | $S, T$ |
| $D$ | $T, S$ | $P, P$ |

(a) Social dilemma payoffs

|   | $C$ | $D$ |
|---|-----|-----|
| $C$ | $-1, -1$ | $-3, 0$ |
| $D$ | $0, -3$ | $-2, -2$ |

(b) PD instance

*multiply payoffs by* $\times 5$

|   | $C$ | $D$ |
|---|-----|-----|
| $C$ | $-1, -1$ | $-3, 0$ |
| $D$ | $0, -3$ | $-2, -2$ |

|   | $C$ | $D$ |
|---|-----|-----|
| $C$ | $-5, -5$ | $-15, 0$ |
| $D$ | $0, -15$ | $-10, -10$ |

(c) Motivating Example

Figure 1: (a) Social dilemma payoff matrix with $R$, $P$, $T$, and $S$. In *Prisoner's Dilemmas (PD)*, $T > R > P > S$ holds (Axelrod, 1984; Macy & Flache, 2002). (b) A PD instance satisfying these inequalities. (c) A motivating example on the reward sensitivity of PI approaches in a 2-player Prisoner's Dilemma: IA requires careful tuning even under the original payoffs and fails to achieve cooperation in both scenarios. LIO achieves high cooperation at the original scale (left) but degrades when the payoffs change, even though the inequalities for greed and fear (Eq. 1) remain valid. In contrast, DRIVE maintains robust cooperation across both settings without retuning.

## 2 Background

**Problem Formulation** In this work, we focus on partially observable *Markov games* (POMGs) $M = \langle \mathbb{D}, \mathbb{S}, \mathbb{A}, \mathbb{P}, \mathbb{U}, \mathbb{Z}, \Omega \rangle$, where $\mathbb{D} = \{1, ..., n\}$ is a set of agents $i$ ($i$ denotes an agent in the decentralized MAS), $\mathbb{S}$ denotes the state space with states $s_t \in \mathbb{S}$ at time step $t$, $\mathbb{A} = \langle \mathbb{A}_1, ..., \mathbb{A}_n \rangle = \langle \mathbb{A}_i \rangle_{i \in \mathbb{D}}$ denotes the tuple of local action spaces, where the corresponding joint action at time $t$ is $a_t = \langle a_{t,i} \rangle_{i \in \mathbb{D}}$, $\mathbb{P}(s_{t+1}|s_t, a_t)$ is the transition probability, $\langle u_{t,i} \rangle_{i \in \mathbb{D}} = \mathbb{U}(s_t, a_t) \in \mathbb{R}$ is the joint reward, $\mathbb{Z}$ is the set of local observations $z_{t,i}$ for each agent $i$, and $\Omega(s_{t+1}) = z_{t+1} = \langle z_{t+1,i} \rangle_{i \in \mathbb{D}} \in \mathbb{Z}^n$ is the subsequent joint observation. Each agent $i$ maintains a local *history* $\tau_{t,i} \in (\mathbb{Z} \times \mathbb{A}_i)^t$. $\pi_i(a_{t,i}|\tau_{t,i})$ denotes the action-selection probability under the individual *policy* of agent $i$. In addition, we assume each agent $i$ to have a *neighborhood* $\mathcal{N}_{t,i} \subseteq \mathbb{D} - \{i\}$ of other agents at every time step $t$, which is domain-dependent, as suggested in (Lupu & Precup, 2020; Yang et al., 2018). $\pi_i$ is evaluated with a *value function* $V_i^\pi(s) = \mathbb{E}_\pi[G_{t,i}|s_t = s]$ for all $s \in \mathbb{S}$, where $G_{t,i} = \sum_{k=0}^\infty \gamma^k u_{t+k,i}$ is the individual discounted *return* of agent $i \in \mathbb{D}$ with discount factor $\gamma \in [0, 1)$ and $\pi = \langle \pi_j \rangle_{j \in \mathbb{D}}$ is the *joint policy* of the MAS. The goal of agent $i$ is to find a *best response* $\pi_i^*$ with $V_i^*(s) = \max_{\pi_i} V_i^{\langle \pi_i, \pi_{-i} \rangle}(s)$ for all $s \in \mathbb{S}$, where $\pi_{-i}$ denotes the joint policy of all agents except $i$. In practice, the global state $s_t$ is not directly observable for any agent $i$, such that $V_i^\pi$ is approximated using local information, i.e., $\tau_{t,i}$ instead (Leibo et al., 2017; Perolat et al., 2017a; Jaderberg et al., 2019). To measure cooperation in the MAS, we define the *social welfare* or *utilitarian metric* $U = \sum_{i \in \mathbb{D}} \sum_{t=0}^{H-1} u_{t,i}$ as the undiscounted sum of rewards.

**Social Dilemmas** *Social dilemmas (SD)* are games where independently optimized policies $\pi_i$ fail to achieve globally optimal outcomes that maximize collective welfare. SDs are commonly studied in 2-player matrix games with two actions: $C$ (cooperate) and $D$ (defect), producing four possible payoffs (Fig. 1): $R$ for mutual cooperation, $P$ for mutual defection, $T$ for exploiting the other, and $S$ for being exploited. A matrix game is a *Prisoner's Dilemma (PD)* if the payoffs satisfy (Axelrod, 1984; Macy & Flache, 2002):

$$T > R > P > S \tag{1}$$

Here, $T > R$ represents *greed*, and $P > S$ *fear*. In *Iterated PDs (IPD)*, an additional condition holds: $2R > T + S$ (Axelrod, 1984; Macy & Flache, 2002). Fig. 1b shows an instance where $D$ is individually rational despite $C$ being socially optimal. As long as the inequalities are satisfied, the game's strategic nature remains invariant to exact payoffs (Axelrod, 1984; Rapoport, 1974). PDs and IPDs are particularly important SDs, as greed and fear often drive agents away from mutual cooperation despite its collective benefit, a phenomenon observed in both nature and human society (Axelrod, 1984; Dawkins, 2016; Rapoport, 1974). *Sequential social dilemmas (SSD)* extend SDs by introducing temporal structure, modeled as stochastic games (Leibo et al., 2017; Perolat et al., 2017b). SSDs enable more realistic scenarios in which behavior is captured by policies rather than by atomic actions. These can still be mapped to matrix games by classifying policies as $C$ or $D$ and evaluating empirical payoffs (Leibo et al., 2017), making core SD concepts applicable to SSDs.

**Multi-Agent Reinforcement Learning** We consider decentralized (independent) learning, where each agent $i$ optimizes its policy $\pi_i$ using local data like $\tau_{t,i}$, $a_{t,i}$, $u_{t,i}$, and $z_{t+1,i}$ via *reinforcement learning (RL)* (Tan, 1993; Foerster et al., 2018; Yang et al., 2018). For methods with peer incentives, the shaped reward replaces $u_{t,i}$ in this trajectory. *Policy gradient RL* is a common method to approximate best responses $\pi_i^*$ (Lowe et al., 2017; Foerster et al., 2018; Yang et al., 2020). A function approximator $\hat{\pi}_{i,\phi_i} \approx \pi_i^*$ is trained using gradient ascent on an estimate of $J = \mathbb{E}_\pi[G_{0,i}]$ (Williams, 1992), where the policy gradient is approximated as (Sutton et al., 2000):

$$g = (G_{t,i} - b_i(s_t))\nabla_{\phi_i} \log \hat{\pi}_{i,\phi_i}(a_{t,i}|\tau_{t,i}) \tag{2}$$

Here, $b_i(s_t)$ is a state-dependent *baseline*, typically approximated by a learned value function $\hat{V}_{i,\omega_i}(\tau_{t,i}) \approx V_i^{\hat{\pi}}(s_t)$ (Foerster et al., 2018). For simplicity, we omit parameters and write $\hat{\pi}_i$, $\hat{V}_i$. Modern actor-critic methods incorporate such variance-reduction baselines, entropy bonuses, or centralized critics, yet still optimize each agent's own return, which remains misaligned with social welfare in social dilemmas. Also, *independent learning* introduces non-stationarity because agents adapt simultaneously (Laurent et al., 2011; Hernandez-Leal et al., 2017), often leading to overly greedy behavior and mutual defection unless rewards or incentives are modified (Leibo et al., 2017; Yang et al., 2020; Phan et al., 2024).

# 3 Dynamic Reward Incentivization

We assume a decentralized MARL setting as formulated in Algorithm 1, where at every time step $t$ each agent $i$ with history $\tau_{t,i}$, policy $\hat{\pi}_i$, and value function $\hat{V}_i$ observes its neighborhood $\mathcal{N}_{t,i}$ through its local observation $z_{t,i}$ and executes an action $a_{t,i} \sim \pi_i(\cdot|\tau_{t,i})$. DRIVE uses $\mathcal{N}_{t,i}$ only for incentive exchange; the underlying policy receives no additional messages beyond the environment observation. The environment then transitions to a new state $s_{t+1} \sim \mathbb{P}(\cdot|s_t, a_t)$ which is observed by each agent $i$ through an observation $z_{t+1,i}$ and a reward $\hat{u}_{t,i}$. The reward is obtained by passing the environmental reward $u_{t,i}$ through an external and possibly unknown *reward-change function* $f_{mod}$ (Alg. 1, l. 11), which simulates modified specifications, varying supply and demand, or sensor degradation (Eysenbach & Levine, 2021; Zhang et al., 2020). In Sec. 5 we analyze an affine special case of $f_{mod}$, applying a shared map varying by epoch. All agents collect their respective *experience tuple* $e_{t,i} = \langle \tau_{t,i}, a_{t,i}, \hat{u}_{t,i}, z_{t+1,i} \rangle$ for PI exchange and independent updates to $\hat{\pi}_i$ and $\hat{V}_i$ (Yang et al., 2020). We assume each agent $i$ has a domain-dependent neighborhood $\mathcal{N}_{t,i} \subseteq D \setminus \{i\}$ at each time step, as introduced in Sec. 2. Lines 13–20 in Alg. 1 correspond to a short parallel communication phase in which neighboring agents exchange DRIVE requests and responses.

---

**Algorithm 1** Multi-Agent Learning with DRIVE

---

1: Initialize parameters of $\hat{\pi}_i$ and $\hat{V}_i$ for all agents $i \in \mathbb{D}$
2: **for** epoch $m \leftarrow 1, E$ **do**
3:      Set $\overline{u}_i \leftarrow 0$ for all agents $i \in \mathbb{D}$          ▷ Reset avg. reward
4:      Sample $s_0$ and set $\tau_{0,i}$ for all agents $i \in \mathbb{D}$
5:      **for** time step $t \leftarrow 0, H-1$ **do**
6:          **for** agent $i \in \mathbb{D}$ **do** $a_{t,i} \sim \hat{\pi}_i(\cdot|\tau_{t,i})$          ▷ Independent decisions
7:          $s_{t+1} \sim \mathbb{P}(\cdot|s_t, a_t)$          ▷ Execute joint action $a_t$
8:          $\langle u_{t,i} \rangle_{i \in \mathbb{D}} \leftarrow \mathbb{U}(s_t, a_t)$          ▷ Environmental rewards
9:          $\langle z_{t+1,i} \rangle_{i \in \mathbb{D}} \leftarrow \Omega(s_{t+1})$
10:         **for** agent $i \in \mathbb{D}$ **do**          ▷ Parallel communication
11:             $\hat{u}_{t,i} \leftarrow f_{mod}(u_{t,i}, m)$          ▷ Ext. reward change
12:             $\overline{u}_i \leftarrow (t \cdot \overline{u}_i + \hat{u}_{t,i})/(t+1)$          ▷ Avg. reward
13:             $u_{t,i}^{DRIVE} \leftarrow DRIVE(\hat{V}_i, \mathcal{N}_{t,i}, \hat{u}_{t,i}, \overline{u}_i)$          ▷ Alg. 2
14:             $e_{t,i} \leftarrow \langle \tau_{t,i}, a_{t,i}, u_{t,i}^{DRIVE}, z_{t+1,i} \rangle$
15:             Update $\tau_{t,i}$ to $\tau_{t+1,i}$ and store $e_{t,i}$
16:         **end for**
17:      **end for**
18:      **for** agent $i \in \mathbb{D}$ **do** Update $\hat{\pi}_i$ and $\hat{V}_i$ using all $e_{t,i}$ of epoch $m$          ▷ Independent updates
19: **end for**

---

**DRIVE Token Exchange** DRIVE uses a reciprocal incentive scheme, inspired by Trivers (1971) and illustrated in Fig. 2a, to exchange dynamic incentives for distributed reward shaping. In the *request phase* (Fig. 2a, 1), each agent $i$ checks its *advantage* or *temporal difference residual* $TD_i(\hat{u}_{t,i})$ (Sutton, 1988):

$$TD_i(\hat{u}_{t,i}) = \hat{u}_{t,i} + \gamma \hat{V}_i(\tau_{t+1,i}) - \hat{V}_i(\tau_{t,i}) \tag{3}$$

If the advantage is non-negative, agent $i$ sends its reward $\hat{u}_{t,i}$ as a *request* to all other agents $j \in \mathcal{N}_{t,i}$. In the *response phase* (Fig. 2a, 2), each request-receiving neighbor $j \in \mathcal{N}_{t,i}$ compares the request $\hat{u}_{t,i}$ to its epoch-average reward $\overline{u}_j$ and computes its response $\Delta_{t,i,j}$ sent from $j$ to $i$, gated by a non-negative TD of $i$:

$$\Delta_{t,i,j} = \begin{cases} \overline{u}_j - \hat{u}_{t,i}, & \text{if } TD_i(\hat{u}_{t,i}) \geq 0, \\ 0, & \text{otherwise.} \end{cases} \tag{4}$$

After this exchange, the DRIVE reward $u_{t,i}^{DRIVE}$ is computed for each agent $i$ as follows:

$$u_{t,i}^{\text{DRIVE}} = \hat{u}_{t,i} - \underbrace{\min\{\langle \Delta_{t,j,i} \rangle_{j \in \mathcal{N}_{t,i}}\}}_{\text{own responses to others' requests}} + \underbrace{\min\{\langle \Delta_{t,i,j} \rangle_{j \in \mathcal{N}_{t,i}}\}}_{\text{others' responses to own requests}}. \tag{5}$$

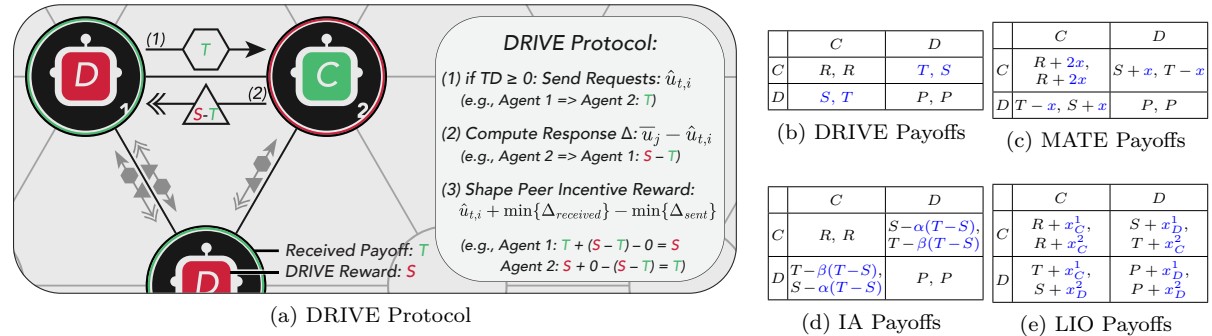

(a) DRIVE Protocol

(b) DRIVE Payoffs

(c) MATE Payoffs

(d) IA Payoffs

(e) LIO Payoffs

Figure 2: DRIVE exchange scheme (a): (1) If $TD_1(\hat{u}_{t,1}) \geq 0$ (Eq. 3), agent 1 sends its reward $\hat{u}_{t,1}$ to neighbor agent 2 as a request, (2) agent 2 calculates the difference $\Delta_{t,1,2}$ between its own average reward $\overline{u}_i$ in the current epoch $m$ and the received request $\hat{u}_{t,i}$, sent back as a response and used to shape the rewards of both agents (Eq. 5). (b)-(e) Modified PD payoff matrices of different PI methods with payoff modifications highlighted in blue (Hughes et al., 2018; Yang et al., 2020; Phan et al., 2024).

Using the responder's epoch-average $\overline{u}_j$ rather than its instantaneous reward makes DRIVE sensitive to systematic exploitation instead of single noisy outcomes: only agents whose recent average return is consistently lower than a neighbor's request generate strong negative $\Delta$ terms. Other combinations (instant–instant, instant–average, average–average) are conceivable and may yield different trade-offs, which is an interesting direction for future work. Appendix A.2 provides an initial conceptual comparison of these variants, and Appendix C.2 complements this discussion empirically.

---

**Algorithm 2** Dynamic Reward Incentive Exchange

1: **procedure** $DRIVE(\hat{V}_i, \mathcal{N}_{t,i}, \hat{u}_{t,i}, \overline{u}_i)$
2:     **if** $TD_i(\hat{u}_{t,i}) \geq 0$ **then** Send request $\hat{u}_{t,i}$ to all $j \in \mathcal{N}_{t,i}$ (Fig. 2a)     ▷ Evaluate acc. to Eq. (3)
3:     **for** neighbor agent $j \in \mathcal{N}_{t,i}$ **if** request $\hat{u}_{t,j}$ received from $j$ **do**     ▷ Handle Requests
4:         $\Delta_{t,j,i} \leftarrow \overline{u}_i - \hat{u}_{t,j}$
5:         $\hat{u}_{req} \leftarrow \min\{\hat{u}_{req}, \Delta_{t,j,i}\}$
6:         Send response $\Delta_{t,j,i}$ to agent $j$ (Fig. 2a)
7:     **end for**
8:     **for** neighbor agent $j \in \mathcal{N}_{t,i}$ **do if** request $\hat{u}_{t,i}$ sent to $j$     ▷ Handle Responses
9:         $\hat{u}_{res} \leftarrow \min\{\hat{u}_{res}, \Delta_{t,i,j}\}$
10:     **end for**
11:     **return** $\hat{u}_{t,i} - \hat{u}_{req} + \hat{u}_{res}$     ▷ Shaped reward (Eq. 5)
12: **end procedure**

---

**Distributed Reward Shaping**   The DRIVE reward $u_{t,i}^{DRIVE}$ is used to update the policies of the corresponding agents using any RL algorithm, e.g., policy gradient methods, according to Eq. (2). Formally, whenever DRIVE is enabled we obtain returns $G_{t,i}$ by replacing $u_{t,i}$ with $u_{t,i}^{\text{DRIVE}}$ in Eq. (2) (cf., Alg. 1, l. 17). Sec. 2 therefore describes the environment-level returns, while DRIVE defines how these are transformed into the shaped rewards that the RL updates optimize. The non-negativity condition of advantage $TD_i(\hat{u}_{t,i})$ in the DRIVE request is needed to expose defecting agents that typically have a greater advantage than the exploited agents in SDs (Axelrod, 1984). If there is unilateral defective behavior, the defective agent $i$ will be penalized by the most exploited neighbor agent $j \in \mathcal{N}_{t,i}$, since $\Delta_{t,i,j} < 0$, which is ensured by the min aggregation terms [1] in Eq. (5). If all agents act equally cooperatively, then all $\Delta_{t,i,j} = 0$, and there is no additional reward or penalty. The complete formulation of DRIVE at time step $t$ for any agent $i$ is given in Algorithm 2. $\hat{V}_i$ is the approximated value function to calculate $TD_i(\hat{u}_{t,i})$, according to Eq. (3), $\mathcal{N}_{t,i}$ is the current local neighborhood, $e_{t,i}$ is the experience tuple, and $\overline{u}_i$ is the current average reward of agent $i$.

---

[1] The min aggregations could be replaced by a sum or mean, similar to (Hughes et al., 2018). However, this would weaken the influence of the most exploited agent relative to others, tolerating individual dissatisfaction that could spread in later epochs.

## 4 Related Work

**MARL in Social Dilemmas**  MARL has achieved substantial progress across a range of domains (Tan, 1993; Littman, 1994; Buşoniu et al., 2010; Vinyals et al., 2019). In decentralized SDs and SSDs, a key challenge is resolving misaligned incentives without centralized control. Recent work addresses this through *local*, peer-induced, or socially conditioned reward-shaping mechanisms (Hughes et al., 2018; Yang et al., 2020; Phan et al., 2024; Altmann et al., 2025), which operate in mixed-motive environments with observation-limited agents and thus align with our setting. Recent approaches also study reciprocity-based intrinsic rewards and decentralized reward mixing to stabilize cooperation in mixed-motive systems (Zhou et al., 2024). By contrast, much of the broader MARL literature, methods for mitigating non-stationarity in cooperative settings (Bowling & Veloso, 2002; Matignon et al., 2007; Wei & Luke, 2016), globally informed reward shaping (Leibo et al., 2017; Devlin et al., 2014), opponent-shaping requiring access to opponents' parameters (Foerster et al., 2018; Letcher et al., 2019; Willi et al., 2022; Zhao et al., 2022), and centralized incentive design (Yang et al., 2022; Guresti et al., 2023) assumes global information or centralized coordination, and therefore does not extend to decentralized SDs.

**Peer Incentivization**  Many PI methods have been introduced to promote mutual cooperation in a distributed fashion via reward exchange (Yi et al., 2022). *Gifting* extends the action space of each agent $i$ with a reward action to incentivize other agents $j \in \mathcal{N}_{t,i}$ (Lupu & Precup, 2020). In contrast, DRIVE is built upon reciprocal reward shaping and does not require an extended action space. Schmid et al. (2021) proposes market-based PI where agents can establish bilateral agreements. A public sanctioning approach, where agents can reward or penalize each other based on known group behavior patterns, has been proposed in (Vinitsky et al., 2023). Hughes et al. (2018) defines an *inequity aversion (IA)* scheme based on non-negative reward differences weighted by two domain-dependent coefficients. *Learning to Incentivize Other learning agents (LIO)* automatically learns an incentive function for each agent $i$ under full observability, based on the joint action of all other agents $j \neq i$ (Yang et al., 2020). Closely related, *Learning to Share* (LToS) (Yi et al., 2022) learns dynamic reward-sharing policies among neighboring agents, enabling decentralized adaptation of incentives. Phan et al. (2022) propose *Mutual Acknowledgment Token Exchange* (MATE), a two-phase protocol where agents exchange fixed tokens $x$: cooperating agents issue tokens while exploited agents receive them, yielding the modified payoff matrix shown in Fig. 2c. Recent work also highlights the brittleness of fixed-token mechanisms under changing reward scales and addresses this limitation by deriving token magnitudes from TD and value estimates and reaching consensus across neighbors, demonstrating robustness to downscaled rewards (Altmann et al., 2025). Most PI methods are sensitive to the reward values of an environment due to relying on fixed incentive values or domain-dependent parameters, thus failing to cooperate when the reward values change (Fig. 1c). Considering the PD as an example, Fig. 2 shows the modified payoff matrices of DRIVE (b), MATE (c), LIO (e), and IA (d). According to these matrices, DRIVE is the only method that does not depend on any particular hyperparameter or explicitly learned value. IA only modifies unilateral cooperation and defection but depends on two domain-dependent coefficients $\alpha$ and $\beta$ (Hughes et al., 2018). LIO learns an incentive function conditioning on the actions of the other agents, which requires sufficient experience and time to adapt accordingly. Similarly, LToS and ME-DIATE introduce adaptive mechanisms for reward sharing or token selection, but their adaptation depends on additional learned policies, value estimates, or consensus updates. In the case of MATE, cooperation depends on an appropriate global token $\mathbf{x} \geq \max\{P - S, \frac{T-R}{3}\}$ that can be derived from the modified payoff matrix. In the example instance in Fig. 1b, $\mathbf{x}$ should be at least 1, which is the original token value used for MATE in (Phan et al., 2024). Interestingly, this value is used for other PI methods as well without further analysis or questioning (Lupu & Precup, 2020; Schmid et al., 2021).

**Teaming**  Beyond independent learners, recent work studies how social preferences, teams, or coalition structures shape incentives in mixed-motive settings, e.g., heterogeneous SVO-based reward shaping, team-based reward sharing, and adaptive reward mixing via price-of-anarchy minimization (Gemp et al., 2022; Radke et al., 2022; McKee et al., 2020; Phan et al., 2021). These methods typically assume fixed or emergent group structures or modify each agent's global utility function. In contrast, DRIVE introduces no explicit coalitions or shared rewards: incentives arise solely through local, bilateral reward exchanges, enabling decentralized alignment without predefined teams or population-level reward mixing.

## 5    Theoretical Analysis

In the following, we analyze the incentive alignment properties of DRIVE in the general PD and its invariance to changing rewards. We emphasize that the mechanism does not guarantee convergence of learning dynamics, but rather reshapes incentives such that cooperation becomes the individually rational choice in repeated interactions. Although our formal discussion focuses on the canonical PD, many well-known sequential social dilemmas (SSDs), such as Coin and Harvest, instantiate PD-like incentive structures, as shown empirically in (Leibo et al., 2017) and explained in the Appendix. In Sec. 8, we further show that DRIVE also yields promising results in more complex SSDs with more than two agents.

**DRIVE in the General Prisoner's Dilemma**   In social dilemmas with a payoff table, as shown in Fig. 1, and inequalities $T > R > P > S$ of Eq. (1), the DRIVE incentives do not change the payoffs $R$ for mutual cooperation and $P$ for mutual defection in the long run because the reward difference $\Delta_{t,i,j}$ would be zero for both agents. With $R > P$, this will still favor mutual cooperation over mutual defection. However, if there were (repeated) unilateral defection, the defective agent $i$ would send a request $T$ to the exploited agent $j$, which responds with a difference of $\Delta_{t,i,j} = S - T < 0$ in the worst case, where the reward of the defective agent $i$ would change to $T + \Delta_{t,i,j} = T + S - T = S$, while the reward of the exploited agent $j$ would change to $S - \Delta_{t,i,j} = S - (S - T) = T$, according to Eq. (5). As shown in Fig. 2b, the DRIVE payoff switch of $T$ and $S$ enables both agents to overcome greed and fear, therefore incentivizing cooperation.

**Theorem 1.** *DRIVE aligns incentives in a generalized two-agent Prisoner's Dilemma such that under TD-gate activation for unilateral defection and sufficiently low epoch-average reward of the exploited agent, mutual cooperation becomes a dominant strategy by effectively reversing the temptation and sucker payoffs.*

*Proof sketch.* As argued above, DRIVE leaves mutual cooperation and mutual defection unchanged in the long run, so the payoffs $\hat{R}$ and $\hat{P}$ are preserved. Under unilateral defection $(D, C)$, Theorem 7 implies that the exploiting agent issues a request, and Lemma 1 yields a strictly negative response $\Delta_{t,i,j} = \overline{u}_j - \hat{T} \in [\hat{S} - \hat{T}, 0)$. Substituting into Eq. (5) gives

$$u_{t,i}^{\text{DRIVE}} = \hat{T} + \Delta_{t,i,j} = \overline{u}_j, \qquad u_{t,j}^{\text{DRIVE}} = \hat{S} - \Delta_{t,i,j}.$$

Hence, unilateral defection is transformed from $(\hat{T}, \hat{S})$ to $(\overline{u}_j, \hat{S} - \Delta_{t,i,j})$, which recovers the payoff inversion $(\hat{T}, \hat{S}) \leftrightarrow (\hat{S}, \hat{T})$ in the steady case $\overline{u}_j \approx \hat{S}$. More generally, whenever $\overline{u}_j < \hat{R}$, defecting against a cooperator yields less payoff than mutual cooperation, so greed is removed while fear remains overcome by $\hat{T} > \hat{P}$. Therefore, under bounded and rational critics, mutual cooperation is the dominant strategy under DRIVE. A full derivation is provided in Appendix A.                                                                              □

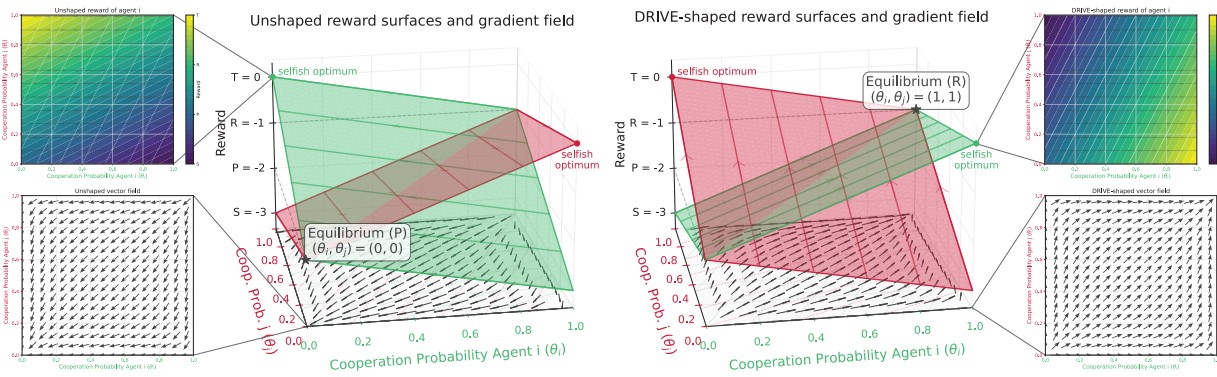

Figure 3: Continuous mean-field view of DRIVE in the 2-player Prisoner's Dilemma. Left: unshaped reward heatmap and induced vector field for agent $i$, showing that local incentives favor defection. Right: DRIVE-shaped reward heatmap and vector field, where the local gradients are redirected toward higher cooperation. Overall, DRIVE resolves greed and fear and turns the dynamics toward mutual cooperation.

Overall, DRIVE does not rely on fixed incentive values or domain-dependent parameters. Thus, it can cope with reward changes that still satisfy the PD inequalities w.r.t. greed and fear, as in the example of Fig. 1c. To make the shaping effect of DRIVE more explicit beyond the discrete payoff tables, Fig. 3 visualizes the corresponding continuous mean-field reward surfaces and optimization dynamics in the 2-player PD. The combined 3D view further connects this continuous picture to the discrete social-dilemma payoffs by showing how shaping resolves the temptation and sucker asymmetries underlying greed and fear. Appendix A.1 provides an extended example that concretely demonstrates the shaping mechanism in a single PD exchange step. Note that Eq. (5) effectively turns the stage game into a coordination game with $(C, C)$ as the unique Nash equilibrium. This reflects standard reward-design practice: instead of relying on learners to infer long-term externalities, we adjust instantaneous payoffs so that cooperation is individually rational while leaving the environment dynamics unchanged.

**Invariance to Changing Rewards** We now analyze DRIVE and MATE with respect to their invariance to changing rewards in the PD, i.e., where the original payoffs $R$, $P$, $T$, and $S$ are dynamically altered by some external non-observable change function $f_{mod}$, to realistically model the assumed drift conditions.

**Definition 1** (Reward Change Function). Let $r_{t,i}$ denote the original environmental reward of agent $i$ at time $t$. The modified reward in epoch $m$ is defined as $r'_{t,i} = f_{mod}(r_{t,i}, m) = c_m r_{t,i} + b_m$, where $c_m > 0$ and $b_m \in \mathbb{R}$ are epoch-dependent scaling and shifting parameters that apply uniformly to all agents and timesteps within epoch $m$. This shared affine map preserves the strategic structure of the game (e.g., PD inequalities) while shifting reward magnitudes and offsets. We focus on this broad but tractable class of per-epoch affine transformations: arbitrary schedules $c_m$ and $b_m$ cover all reward dynamics in Sec. 8 and remain compatible with our per-epoch normalization scheme. Further illustrations and edge cases, including epochs where $c_m \approx 0$, are provided in Appendix A.8.

**Theorem 2.** *Fixed-token MATE agents are not invariant to rewards altered by $f_{mod}$ in the general PD.*

*Proof.* According to the MATE payoff matrix in Fig. 2c, the global token $\mathbf{x} = x_1 = x_2 > 0$ must satisfy $\mathbf{x} \geq \max\{P - S, \frac{T-R}{3}\}$ for emergent cooperation. When the change function $f_{mod}$ is chosen such that $f_{mod}(P - S) > \mathbf{x}$ or $f_{mod}(\frac{T-R}{3}) > \mathbf{x}$, then MATE agents are no longer guaranteed to cooperate mutually. $\square$

Note that Theorem 2 can be generalized to any other PI method that uses fixed peer incentive values. One could, in principle, re-tune $x$ after each change in $f_{mod}$, but this requires additional global knowledge or meta-optimization, whereas DRIVE adapts automatically through local reward differences.

**Theorem 3.** *DRIVE agents are invariant to changing rewards in the general PD, altered by $f_{mod}$ .*

*Proof.* Altering the original payoffs $R$, $P$, $T$, and $S$ by any $f_{mod}$ would proportionally change the DRIVE payoff matrix in Fig. 2b. Thus, DRIVE agents are always incentivized to cooperate mutually as the inequalities to overcome greed and fear remain satisfied. $\square$

We now extend the invariance analysis from the PD to general SSDs using policy gradient methods with return normalization, as detailed in Appendix A.10.

**Theorem 4.** *DRIVE agents are invariant to changing rewards in SSDs, where environmental rewards are altered by $f_{mod}$ (Definition 1), when trained with normalized policy gradient methods (Lemma 7).*

Consider sequential social dilemmas with returns normalized per epoch to zero mean and unit variance. If rewards are transformed by a positive affine map per epoch, then both (i) standard policy gradient learning and (ii) DRIVE's shaped rewards remain invariant with respect to the normalized return $\tilde{G}_{t,i}$. Consequently, the incentive alignment induced by DRIVE persists under per-epoch reward scaling and shifting within this class of transformations. Overall, under per-epoch normalization, both the base policy-gradient updates and DRIVE's shaped rewards are invariant to shared per-epoch positive affine reward transformations (Lemma 7).

*Proof.* The result follows from two components. First, Lemma 7 shows that standard policy gradient RL is invariant to positive affine reward transformations under continuing discounted returns and per-epoch normalization. Second, DRIVE shapes rewards via differences between epoch-average and instantaneous rewards. Applying $f_{mod}$ gives $\Delta'_{t,i,j} = f_{mod}(\overline{u}_i) - f_{mod}(u_{t,i}) = f_{mod}(\Delta_{t,i,j})$. Hence the DRIVE-shaped reward also transforms affinely: $u_{t,i}^{DRIVE'} = c_m u_{t,i}^{DRIVE} + b_m$. Applying Lemma 7 to the resulting continuing discounted returns yields identical normalized returns. Therefore, both the base policy-gradient updates and DRIVE's shaped rewards are invariant to such transformations. $\square$

**Robustness to noise perturbations** Definition 1 considers shared affine transformations and denotes the resulting rewards by $\hat{r}$. We now consider a different source of variation: stochastic reward perturbations.

**Definition 2** (Noisy reward perturbation). Let $r_{t,i}$ denote the environmental reward of agent $i$ at time $t$. The perturbed reward in epoch $m$ is $\tilde{r}_{t,i} = r_{t,i} + \varepsilon_{m,i}$, with independent Gaussian noise $\varepsilon_{m,i} \sim \mathcal{N}(0, \sigma^2)$. Unless stated otherwise, all perturbations are mutually independent and share the same variance $\sigma^2$.

Assume equally spaced PD payoffs $T > R > P > S$, with $T-R = R-P = P-S = d$. The DRIVE mechanism in Theorem 1 depends on preserving the ordering between exploitation and cooperation incentives. For two independently perturbed rewards with gap $D$, where $\Phi$ denotes the standard normal CDF, Lemma 8 gives:

$$\Pr[\tilde{r} > \tilde{r}'] = \Phi\left(\frac{D}{\sqrt{2}\sigma}\right) \tag{6}$$

**Theorem 5** (Robustness under per-agent perturbations). *Under per-agent reward perturbations according to Definition 2, DRIVE aligns incentives according to Theorem 1 with high probability ($\gtrsim 0.999$) for $\sigma \le d/\sqrt{2}$.*

*Proof.* Assume that each agent receives one shared perturbation across all payoff entries. Then the local payoff ordering $T > R > P > S$ remains unchanged. The only remaining failure mode is a cross-agent inversion such as $\tilde{T}_i \le \tilde{S}_j$. Substituting the relevant gap is $D = T - S = 3d$ into Eq. (6) and choosing $\sigma_w = d/\sqrt{2}$ hence preserves the exploitation ordering with probability $\Pr[\tilde{T}_i > \tilde{S}_j] = \Phi(3) \approx 0.999$. $\square$

**Theorem 6** (Robustness under independent payoff perturbations). *For independent payoff perturbations, even local PD inequalities may change. In this stricter setting, preserving adjacent payoff orderings such as $R > P$ with the same confidence requires $\sigma \le d/3\sqrt{2}$.*

*Proof.* Assume that each payoff entry is perturbed with independent $\varepsilon. \sim \mathcal{N}(0, \sigma^2)$, s.t. $\tilde{T} = T + \varepsilon_T$, $\tilde{R} = R + \varepsilon_R$, $\tilde{P} = P + \varepsilon_P$, $\tilde{S} = S + \varepsilon_S$. To preserve adjacent local PD payoff inequalities such as $R > P$ with probability $\Phi(3) \approx 0.999$, using the corresponding payoff gap $D = R - P = d$ with Eq. (6) gives $\sigma_n = d/3\sqrt{2}$. $\square$

Consequently, DRIVE remains robust under comparatively large per-agent perturbations and degrades gracefully under fully independent payoff perturbations as long as the underlying payoff ordering is preserved with sufficiently high probability. Figure 4 visualizes how increasing perturbation scales induce overlap between payoff distributions and thereby reduce the probability that the strategic ordering is preserved. Appendices A.12 and C.1 provide further details, a complete ordering analysis over the induced payoff permutations, and preliminary empirical results evaluating the robustness to per-agent noise perturbations.

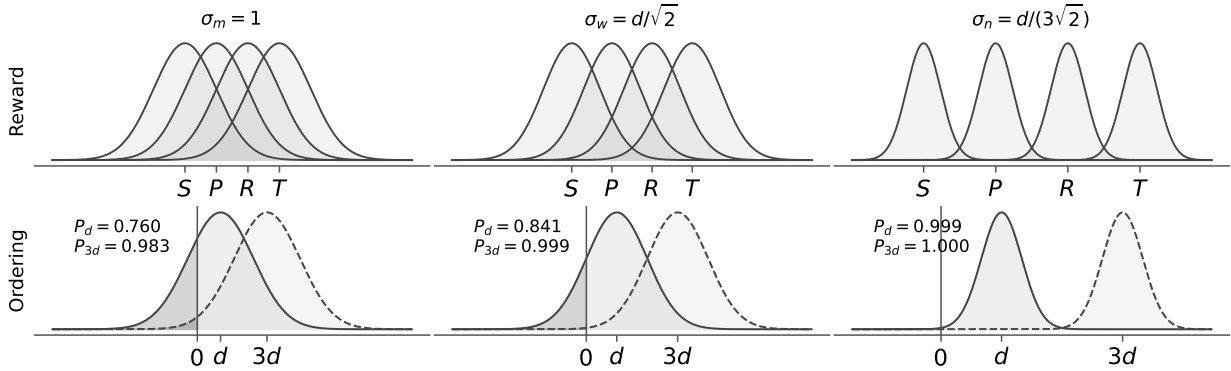

Figure 4: Noise-induced ordering changes under equal payoff spacing $T - R = R - P = P - S = d$. The top row visualizes the overlap between noisy payoff distributions for increasing perturbation scales. The bottom row shows the corresponding difference distributions governing the ordering probability $\Pr[\tilde{r} > \tilde{r}']$. The wider scale $\sigma_w = d/\sqrt{2}$ preserves the exploitation ordering $T > S$ with probability $\Phi(3) \approx 0.999$, while the narrower scale $\sigma_n = d/(3\sqrt{2})$ preserves adjacent payoff orderings such as $R > P$ with the same confidence.

**Effects of Non-Adherence to the Protocol**   The preceding analysis assumes that all agents truthfully and synchronously follow the DRIVE protocol, i.e., each agent shares its epoch-average reward, responds to requests, and applies the shaping rule consistently. In practice, however, communication failures, delays, or strategic misreporting may occur. Here we theoretically analyze the impact of such *non-adherence* compared to other PI approaches; for a detailed example, see Appendix A.5.

- **Full compliance** When all agents follow the protocol truthfully, the results of Theorem 1 and Theorems 3–4 apply. Payoffs $T$ and $S$ are swapped under unilateral defection, ensuring incentive alignment toward cooperation. This reasoning can be generalized to SSDs beyond the PD, provided the characteristic inequalities for greed and fear are preserved.

- **No requests sent (despite TD $\geq 0$)** An agent that withholds requests can still receive requests and respond truthfully. For its neighbors, this means one fewer reciprocal request. Due to the min aggregation, their penalization levels may increase slightly, but are not substantially affected as long as other agents send requests. For the non-requesting agent itself, $\hat{u}_{res} = 0$, so it forfeits the opportunity for reciprocal improvement and is indirectly penalized. If the agent defects, it may temporarily avoid penalization, but overall, it loses the benefits of mutual shaping and does not gain a stable advantage.

- **No responses sent** If an agent does not respond to requests (due to communication loss or intentional withholding), the effect depends on the context: If the agent is defecting, withholding its negative $\Delta$ weakens the penalty on its exploited neighbor. However, it also loses the chance to benefit from reciprocity, so deliberate non-response is not rational. If the agent is cooperative, the missing (typically positive or small) $\Delta$ has a limited effect because the min aggregation dampens single missing contributions. Thus, non-response is mostly neutral or slightly beneficial for defectors but costly overall.

- **False requests or responses** Two types of misbehavior are possible: *Sending requests with TD < 0:* This can occur due to noise or early training misestimation. Such requests typically yield small or positive differences that are filtered out by the min operator and thus have little impact. *Sending false responses:* An agent could deliberately misreport $\Delta$ to distort shaping. In the worst case, the min aggregation could impose unjustified penalties on requesters, potentially undermining cooperation. This depicts a fragility and limitation of DRIVE.

- **No compliance at all** If all agents withhold or communication fails globally, then all shaping terms vanish. The system collapses to plain MARL without incentives, and mutual defection becomes a possible equilibrium again. This is similar to fixed-token methods like MATE, which also break down if their coordination mechanism is unavailable.

**Conceptual Discussion**   DRIVE offers a simple, adaptive PI mechanism for fostering cooperation in SDs with changing rewards. It operates through local peer interactions, requiring no central controller, global knowledge, or all-to-all communication. At the same time, DRIVE relies on truthful peer communication to function correctly. Partial compliance leads to *graceful degradation*: as long as at least one honest responder provides feedback, defectors are still penalized; if all agents fail to comply, DRIVE reduces to baseline MARL dynamics. Empirical robustness of reciprocal token exchange under anomalous protocol deviations and communication failures has been studied in Phan et al. (2024). Compared to fixed-token methods, DRIVE is at least as robust and often more adaptive, but its reliance on strict adherence to the exchange protocol introduces a new potential fragility. Designing protocol variants that are robust to partial compliance – for example, through redundant aggregation rules, stochastic auditing, or alternative shaping operators– remains an important avenue for future work. As shown in the modified PD payoff tables in Fig. 2b, DRIVE only depends on the environmental rewards without requiring fixed incentive values $x$ as MATE, domain-dependent coefficients $\alpha, \beta$ as IA, or time-consuming incentive learning of $x_C^i, x_D^i$ as LIO.

**Scalability**   Similar to (Hughes et al., 2018; Yang et al., 2020; Phan et al., 2024), Algorithm 2 scales linearly with $\mathcal{O}(4(n-1))$ in the worst case with respect to incentive exchanges. Note that given the characteristics of greed and fear exhibited by general PDs, the above theoretical analysis generally applies to various SDs. However, the core incentive-alignment result is derived for the clean two-agent PD, while SSDs such as Coin and Harvest introduce longer horizons, non-stationarity, temporal credit assignment, and delayed effects of incentive exchange. Appendix B provides details on how the used Coin and Harvest environments can be

converted into matrix-game abstractions similar to the IPD, showing that the simplified strategic structures satisfy the required dilemma conditions. Thus, DRIVE can be used in any SD, where rewards change dynamically through an external and unknown change function $f_{mod}$. If the conditions for cooperation, i.e., characteristic inequalities of the SD, remain the same, DRIVE can adapt and maintain cooperation, as shown in Fig. 1c and later demonstrated. However, a complete convergence proof for the full actor–critic learning dynamics in these temporally extended SSDs is beyond the present analysis. Finally, a key modeling assumption is the neighborhood $N_{t,i}$ that restricts which agents can exchange incentives. In our experiments, this neighborhood is tied to the environment as introduced in the subsequent section. Crucially, DRIVE's min-aggregation means that a defector's shaped reward depends on its most exploited neighbor, so, unlike simple reward-sharing schemes, penalties do not dilute with larger populations. Appendix A.7 generalizes the 2-agent result to graphical $N$-agent PDs, showing that pairwise incentive alignment propagates through sufficiently connected communication graphs. Thus, cooperation incentives degrade smoothly rather than collapsing abruptly under partial compliance. In particular, defector penalization is guaranteed when compliant agents form a dominating set, i.e., when every potential requester has at least one compliant neighbor.

# 6 Experimental Setup[2]

We use three well-known SSDs based on (Foerster et al., 2018; Perolat et al., 2017a). At every time step, the order of agent actions is randomized to resolve conflicts, e.g., when multiple agents step on a coin or tag each other simultaneously. All SSDs represent PD instances, as empirically shown in (Leibo et al., 2017) and Appendix B. Since we modify the rewards by $f_{mod}$, we assess the cooperation with domain-specific measures. For each experiment, all respective algorithms were run 20 times over $E = 4,000$ epochs of 10 episodes to report the average progress and the 95% confidence interval.

**Iterated Prisoner's Dilemma** For IPD, we use the payoff matrix shown in Fig. 1b. Both agents observe the previous joint action $z_{t,i} = a_{t-1}$ at every time step $t$, which is the zero vector at the start state $s_0$. The Nash equilibrium is always to defect (DD). An episode consists of $H = 150$ iterations, and we set $\gamma = 0.95$. The neighborhood $\mathcal{N}_{t,i} = \{j\}$ is defined by the other agent $j \neq i$. We measure the *Cooperation Rate* as the cooperation (CC) count per episode divided by $H$.

**Coin** *Coin-2* and *Coin-4* are SSDs and consist of $n \in \{2, 4\}$ agents with different colors, which start at random positions and have to collect a coin with a random color and position (Lerer & Peysakhovich, 2017; Foerster et al., 2018). If an agent collects a coin, it receives a reward of $+1$. However, if the coin has a different color than the collecting agent, another agent with the actual matching color is penalized with -2. After being collected, the coin respawns randomly with a new color. All agents can observe the entire field and can move north, south, west, and east. Each agent can only determine whether a coin is the same color as itself. An episode terminates after $H = 150$ time steps, and we set $\gamma = 0.95$. The neighborhood $\mathcal{N}_{t,i} = \mathbb{D} - \{i\}$ is defined by all other agents $j \neq i$. We measure the *own coin rate* $P(own\ coin) = \frac{\#\ collected\ coins\ with\ same\ color}{\#\ all\ collected\ coins}$ based on the coins collected by each agent.

**Harvest** *Harvest-12* is an SSD and consists of $n = 12$ agents, starting at random positions and having to collect apples. The apple regrowth rate depends on the number of surrounding apples (Perolat et al., 2017a). If all apples are harvested, no apples will grow until the episode terminates. At every time step, all agents receive a time penalty of -0.01. For each collected apple, an agent receives a reward of $+1$. All agents have a $7 \times 7$ field of view and can do nothing, move north, south, west, or east, or tag other agents within their field of view with a tag beam of width 5, pointed to a specific cardinal direction. If an agent is tagged, it is unable to act for 25 time steps. Tagging does not yield any rewards. An episode terminates after $H = 250$ time steps, and we set $\gamma = 0.99$. The neighborhood $\mathcal{N}_{t,i}$ is defined by all other agents $j \neq i$ being in the field of view of agent $i$. We measure the *Sustainability*, defined by the average number of time steps at which apples are collected.

---

[2]Implementations are appended and will be open-sourced upon publication.

**MARL Algorithms** To isolate the impact of reward changes on PI mechanisms rather than the base RL algorithm, we implement all methods, including DRIVE, using the same standard policy-gradient backbone (Eq. 2) with normalized returns (Appendix A.10), following (Hessel et al., 2019). This shared backbone also serves as the Naive Learning baseline without any PI-based reward shaping (Foerster et al., 2018). We use LIO, MATE with $x = 1$ as suggested in (Phan et al., 2024), and IA with $\alpha = 5$ and $\beta = 0.05$ as representative state-of-the-art PI baselines (Hughes et al., 2018; Yang et al., 2020; Phan et al., 2024)[3]. For *IPD* and *Coin-2*, we directly include the reported performance of the opponent-shaping techniques LOLA-PG and POLA-DiCE from (Foerster et al., 2018; Zhao et al., 2022), due to the high computational cost of second-order derivative calculations.

## 7 Experimental Results under Stationary Rewards

**Setting** We compare DRIVE with the baselines in *IPD*, *Coin-2*, *Coin-4*, and *Harvest-12* without any reward change, i.e., $f_{mod}(u, m) = u$ (Algorithm 1, Line 15), in all epochs $m$.

**Results** The results are shown in Fig. 5. DRIVE achieves competitive and stable cooperation in *IPD* compared with all baselines. In *Coin-2*, DRIVE and MATE achieve the highest *own coin* rate, significantly outperforming POLA-DiCE, which is more cooperative than LIO, IA, and LOLA-PG. In both larger SSDs, namely *Coin-4* and *Harvest-12*, DRIVE and MATE achieve the highest level of cooperation, where DRIVE is slightly more cooperative in *Coin-4*, while MATE is slightly more cooperative in *Harvest-12*.

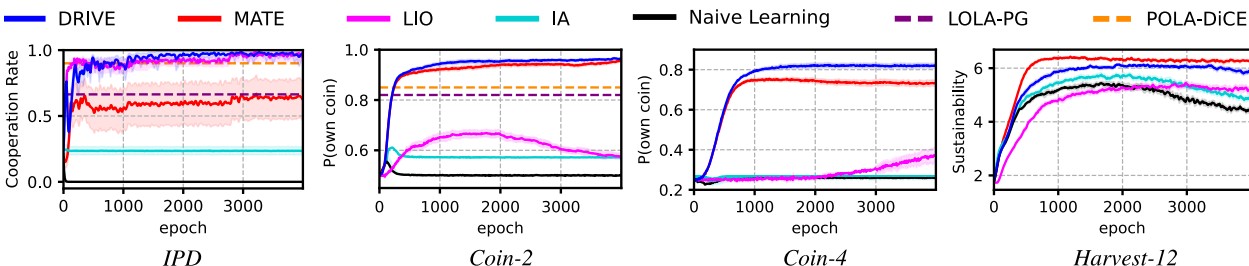

Figure 5: Average progress of DRIVE and other baselines in SSDs without reward change. Shaded areas show the 95% confidence interval. The results of LOLA-PG and POLA-DiCE are from (Foerster et al., 2018; Zhao et al., 2022).

## 8 Experimental Results under Dynamic Reward Changes

**Setting** Next, we evaluate DRIVE and the baselines with different reward change functions $f_{mod}$.

I **Linear increase:** $f^I_{mod}(u, m) = u(\eta m + 1)$. This models a steadily increasing reward scale, for instance, due to rising system-wide demand or progressively stronger performance targets.

II **Exponential decay:** $f^{II}_{mod}(u, m) = ue^{-\eta m}$. This models gradual reward attenuation, e.g., from sensor wear, hardware degradation, or slowly worsening signal quality over time.

III **Stepwise increase:** $f^{III}_{mod}(u, m) = u(\lfloor \eta m \rfloor + \chi)$. This models discrete distribution shifts with abrupt rescalings, as commonly observed when switching between training phases, simulators, or simulation-to-reality deployment.

IV **Damped cosine modulation:** $f^{IV}_{mod}(u, m) = \eta + u(1 - \frac{m}{E}) \cos^2(2\eta m)$. This models irregular, non-monotonic reward changes (with a decaying amplitude), for example from evolving specifications, intermittent human feedback, or online reward redesign.

---

[3]The hyperparameters follow prior studies that tuned the methods on the same or comparable environments. We thus test the robustness of fixed-incentive cooperation mechanisms under reward-scale changes, where DRIVE requires no re-tuning.

We set $\eta = 0.001$ and $\chi = 10$. Together, these functions cover monotone growth (I, III), convergence toward zero (II), and oscillatory, time-varying shifts (IV), while preserving the underlying payoff ordering of the social dilemma.

**Results** The results are shown in Fig. 6. The learning curve of DRIVE is not significantly affected by training with the functions $f_{mod}^I$, $f_{mod}^{II}$, and $f_{mod}^{III}$. Function $f_{mod}^{IV}$ causes occasional dips in the learning curve of DRIVE, where the scaling factor approaches zero, temporarily collapsing the payoff ordering of the underlying dilemma. However, since these low-scale epochs are isolated and the modulation quickly moves away from this regime, DRIVE rapidly recovers its learning dynamics and performance level. LIO is robust against any change function in *Coin-4*, but its cooperation level significantly deteriorates in other domains. Despite automatically learning an incentive function, LIO is especially sensitive to $f_{mod}^{IV}$. IA is affected in *Coin-2* and *Coin-4*, where it never outperforms Naive Learning. MATE is only able to resist reward changes in *Harvest-12* when training with $f_{mod}^I$. In all other cases, MATE's cooperation level significantly deteriorates, even worse than Naive Learning in some cases. The learned behavior is least stable with $f^{IV}$, where MATE is not able to recover from dips in the learning curve, unlike DRIVE. DRIVE is the only PI method that outperforms Naive Learning, LOLA-PG, and POLA-DiCE in *IPD* and *Coin-2*. Appendix C provides further cooperation metrics (social welfare, equality, sustainability, and peace) in *Harvest-12*.

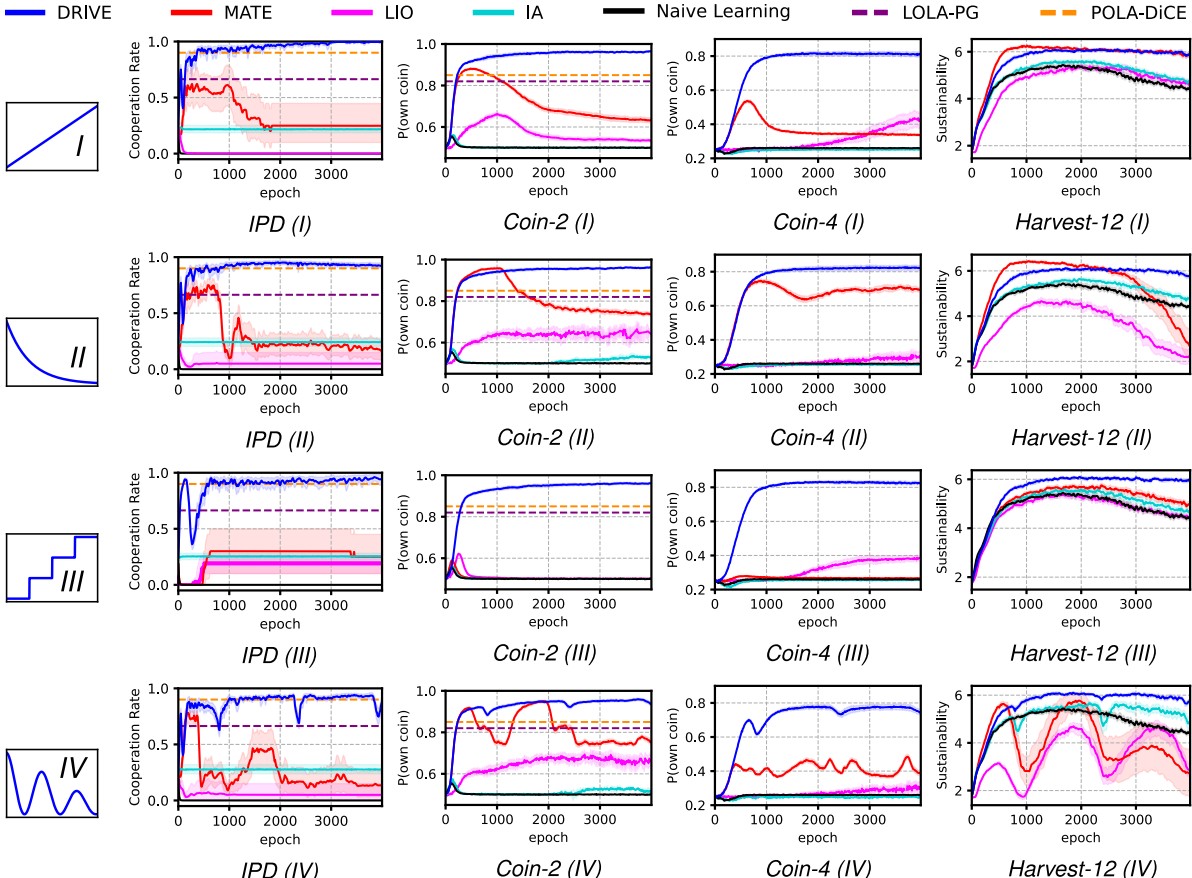

Figure 6: Average progress of DRIVE and other baselines in SSDs with different reward change functions $f_{mod}$ I, II, III, and IV, illustrated left and in Fig. 12. Shaded areas show the 95% confidence interval. The results of LOLA-PG and POLA-DiCE are from (Foerster et al., 2018; Zhao et al., 2022). The non-PI approaches Naive Learning, LOLA-PG, and POLA-DiCE are largely insensitive to the reward changes of $f_{mod}$ considered here due to per-epoch return normalization, as explained in Appendix A.10.

## 9    Discussion

We presented DRIVE, an adaptive PI approach that achieves and maintains cooperation in SDs with changing rewards. DRIVE agents reciprocally exchange reward differences to incentivize mutual cooperation in a fully decentralized manner. Our theoretical analysis shows how DRIVE aligns incentives toward mutual cooperation in the general Prisoner's Dilemma by overcoming greed and fear while remaining invariant to per-epoch reward scaling, and empirically robust to the considered reward changes. This contrasts with prior PI methods that rely on fixed incentive values, domain-dependent coefficients, or time-consuming incentive learning. Consequently, DRIVE is better aligned with game-theoretic assumptions and safer to deploy in environments with drifting rewards. Our experiments further demonstrate that DRIVE significantly outperforms state-of-the-art methods such as LIO, MATE, and IA, as well as opponent-shaping baselines such as POLA-DiCE and LOLA-PG, across multiple SSDs with dynamic reward changes. The results confirm the sensitivity of existing PI approaches to reward magnitudes and highlight DRIVE's ability to sustain cooperation under such transformations, provided the underlying strategic inequalities remain intact.

**Limitations**   Despite promising results, DRIVE relies on truthful and synchronous peer communication. Partial non-compliance results in graceful degradation of performance. However, the current mechanism does not enforce that agents report rewards or reward differences honestly. Strategic misreporting may therefore distort the min aggregation and lead to missing or unjustified penalties, especially when compliant responders are sparse. Moreover, the current theoretical analysis assumes homogeneous populations and complete or sufficiently well-connected communication graphs. These assumptions may not hold in more realistic settings with strategic manipulation, communication noise, heterogeneous agents, or sparse interaction structures. Furthermore, the present theoretical guarantees should be interpreted primarily as incentive-alignment results, with convergence guarantees currently limited to simplified repeated-game abstractions. The core proof is derived for the clean two-agent PD, where greed and fear can be analyzed directly through the payoff inequalities. Appendix A.6 complements this discrete payoff analysis with a continuous mean-field view in which policies are parameterized by cooperation probabilities; in this setting, DRIVE reshapes the local reward landscape such that the induced gradient field points toward mutual cooperation. Appendix A.7 extends the pairwise incentive-alignment argument to graphical $N$-agent PDs, and Appendix B shows how the empirical Coin and Harvest environments can be abstracted into matrix games similar to the IPD. For these simplified abstractions, the required dilemma structure is preserved. However, these results still do not constitute a convergence proof for the full actor–critic learning dynamics used in the temporally extended SSD experiments. The actual SSDs additionally introduce longer horizons, non-stationarity from simultaneous learning, temporal credit assignment, critic approximation error, and delayed alignment between an action, the resulting reward difference, and the subsequent incentive exchange. These factors can increase the variance of individual returns and may affect the speed or stability with which cooperative policies emerge.

**Future Work**   Several directions for future work follow from these observations. This includes studying how DRIVE interacts with high-variance actor–critic learning dynamics, whether return variance affects convergence speed, and under which assumptions the temporal alignment between actions, reward differences, and incentive exchanges can be guaranteed. Extending the analysis beyond PD-like inequalities to broader classes of SDs would further clarify when DRIVE can be expected to induce cooperation. A second direction is to develop mechanisms that are robust to partial or strategic non-compliance, for example, through redundant aggregation, stochastic auditing, or consensus systems (Altmann et al., 2025), as well as explicit incentives for truthful reporting. Such safeguards could help close the gap between the idealized protocol analyzed here and open multi-agent deployments with unreliable or adversarial peers. Extending the theoretical analysis to heterogeneous populations and more general network topologies represents another important step toward real-world applicability. A natural next step is to study learned and dynamic neighborhoods, and to combine DRIVE with models of evolving social structure and coalitions, as suggested by recent work on reward sharing in teams and coalitions. Finally, incorporating agent identification could enable targeted bilateral responses, e.g., against adversarial peers, and improve robustness under sparse or changing communication graphs.

## Broader Impact Statement

This work proposes a decentralized peer-incentivization mechanism designed to promote cooperation in multi-agent systems under changing rewards. A key positive impact is improved robustness of cooperative behavior when reward magnitudes vary over time, which is relevant for real-world applications such as distributed control, resource management, and multi-robot coordination. By reducing dependence on fixed incentive magnitudes, the approach may help bridge gaps between training and deployment conditions. At the same time, the method relies on inter-agent communication and protocol adherence. In practice, agents may misreport or fail to communicate, which could degrade cooperation or introduce vulnerabilities. These risks are particularly relevant in complex or human-facing environments not captured by the settings studied here. We therefore view this work as a step toward more adaptive decentralized cooperation, but not as a complete solution for real-world deployment. Future work should investigate robustness to non-compliance and adversarial behavior, as well as safeguards such as auditing or constraints on incentive exchange.

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

# A  Extended Theoretical Results

In this section, we provide additional worked-out examples and detailed arguments to complement the theoretical analysis in the main body. These results give a more concrete and intuitive understanding of how DRIVE operates in simple settings, how its behavior changes under different assumptions of protocol adherence, and how the microscopic request/response mechanism underpins the macroscopic equilibrium results, and provides a simple extension to larger multi-agent settings.

**Preliminaries**  We recall the standard two-player Prisoner's Dilemma (PD) with actions $C$ (cooperate) and $D$ (defect) and payoffs $(R, R)$, $(P, P)$, $(T, S)$, and $(S, T)$ for profile $(C, C)$, $(D, D)$, $(D, C)$, and $(C, D)$, respectively (Fig. 1). The game is a PD if $T > R > P > S$ (cf. Eq. 1), where $T > R$ encodes greed and $P > S$ encodes fear (Axelrod, 1984). We denote the environmental reward of agent $i$ at time $t$ by $u_{t,i}$, the externally modified reward in epoch $m$ by $\hat{u}_{t,i} = f_{mod}(u_{t,i}, m)$, and the running average of $\hat{u}_{t,i}$ within epoch $m$ by $\bar{u}_i$. The DRIVE shaping rule for agent $i$ at time $t$ is

$$u_{t,i}^{\text{DRIVE}} = \hat{u}_{t,i} - \min\{\langle \Delta_{t,j,i} \rangle_{j \in \mathcal{N}_{t,i}}\} + \min\{\langle \Delta_{t,i,j} \rangle_{j \in \mathcal{N}_{t,i}}\}, \quad \text{with} \tag{7}$$

$$\Delta_{t,i,j} = \begin{cases} \bar{u}_j - \hat{u}_{t,i}, & \text{if } TD_i(\hat{u}_{t,i}) \geq 0, \\ 0, & \text{otherwise,} \end{cases} \tag{8}$$

where $\Delta_{t,j,i}$ is the *response that $i$ sends* to $j$'s request (hence it is *subtracted*), while $\Delta_{t,i,j}$ is the *response received* by $i$ to its own request (hence it is *added*). Intuitively, $TD_i(\hat{u}_{t,i}) \geq 0$ means that the realized one-step outcome was at least as good as what the critic predicted.

## A.1  Single Exchange Mechanism Underlying Incentive Alignment in a 2-Player Prisoner's Dilemma

We give a concrete, single-step derivation showing how the DRIVE request/response protocol (Alg. 2) together with the TD gate in Eq. (3) and the shaping rule in Eq. (4) yields the *payoff swap* $T \leftrightarrow S$ under unilateral defection. This turns the PD into a coordination game with $(C, C)$ as the unique Nash equilibrium. Unless otherwise stated, we assume full compliance with the protocol.

**Setup**  Consider a single time step $t$ in a 2-agent PD with payoffs $T > R > P > S$ and neighborhoods $N_{t,1} = \{2\}$, $N_{t,2} = \{1\}$. Rewards may already be transformed by $f_{mod}$ in epoch $m$; for notational convenience we write

$$\hat{T} := f_{mod}(T, m), \quad \hat{R} := f_{mod}(R, m), \quad \hat{P} := f_{mod}(P, m), \quad \hat{S} := f_{mod}(S, m).$$

We assume that, in the long run under stationary play, epoch averages coincide with the corresponding instantaneous payoffs for each action profile (e.g., under $(C, C)$ both agents have $\bar{u}_i = \hat{R}$). Concretely, we assume $\bar{u} \approx \hat{u}$ in the following to isolate the intended mechanism; the subsequent discussion explains how deviations $\bar{u}_j \neq \hat{u}_{t,j}$ do not fundamentally change the direction of shaping in exploitation states, but modulate its magnitude to reflect longer-term (epoch-level) advantage.

**Mutual cooperation and mutual defection**  If both agents cooperate at time $t$, each receives $\hat{u}_{t,1} = \hat{u}_{t,2} = \hat{R}$ and, in steady state, $\bar{u}_1 = \bar{u}_2 = \hat{R}$. Hence $\Delta_{t,1,2} = \bar{u}_2 - \hat{u}_{t,1} = 0$ and $\Delta_{t,2,1} = 0$, so Eq. (7) yields

$$u_{t,1}^{\text{DRIVE}} = \hat{R}, \quad u_{t,2}^{\text{DRIVE}} = \hat{R}.$$

A similar argument shows that under mutual defection, both agents retain payoff $\hat{P}$. Thus, DRIVE leaves $R$ and $P$ unchanged in the long run.

**Unilateral defection**  Now consider a state where agent 1 defects ($D$) and agent 2 cooperates ($C$), so the instantaneous modified rewards are $\hat{u}_{t,1} = \hat{T}$, $\hat{u}_{t,2} = \hat{S}$. Under the usual PD ordering, the defector benefits from a higher reward than the cooperator. We assume that this yields a non-negative TD advantage for the defector and a non-positive one for the cooperator, so that

$$TD_1(\hat{u}_{t,1}) \geq 0 \Rightarrow \text{agent 1 sends a request}, \quad TD_2(\hat{u}_{t,2}) < 0 \Rightarrow \text{agent 2 does not send.} \tag{9}$$

This corresponds to the intended operating regime of DRIVE and matches the empirical behavior observed in our experiments. Agent 1 sends its instantaneous reward $\hat{u}_{t,1} = \hat{T}$ to agent 2 as a request. Upon receiving this, agent 2 computes the (epoch-$m$) difference $\Delta_{t,1,2} = \overline{u}_2 - \hat{u}_{t,1}$. In the PD step considered, the worst-case consistent choice is $\overline{u}_2 = \hat{S}$, yielding $\Delta_{t,1,2} = \hat{S} - \hat{T} < 0$. Agent 2 returns this value to agent 1. Because only agent 1 sent a request, the reverse difference $\Delta_{t,2,1}$ is undefined in this step; by Algorithm 2 it defaults to 0 for aggregation when no responses arrive. Using the responder's epoch-average $\overline{u}_j$ rather than its instantaneous reward makes DRIVE sensitive to systematic exploitation instead of single noisy outcomes. Only agents whose recent average return is consistently lower than a neighbor's request generate strong negative $\Delta$ terms. If both sides used instantaneous rewards, a single lucky or unlucky step could trigger large transfers, making incentives noisy and easy to game via short-term risk-taking. Using instantaneous rewards against averages would instead turn DRIVE into a risk-sharing mechanism, while comparing averages on both sides would be very stable but slow to react. The chosen average–instant design therefore balances robustness to noise with responsiveness to emerging exploitation. Moreover, the sign pattern in exploitation states is stable: since $\hat{T}$ is the maximal one-step payoff in the PD, any bounded epoch-average $\overline{u}_2 \leq \hat{T}$ implies $\Delta_{t,1,2} = \overline{u}_2 - \hat{T} \leq 0$, so the response received by the defector remains non-positive even when $\overline{u}_2 \neq \hat{S}$. Deviations $\overline{u}_2 > \hat{S}$ soften the swap (smaller penalty), while $\overline{u}_2 < \hat{S}$ strengthens it (larger penalty), reflecting epoch-level disadvantage. In the 2-player case, the minima over neighbors reduce to single terms. The per-step shaped rewards are

$$\hat{u}_{t,1}^{\mathrm{DRIVE}} = \hat{u}_{t,1} - \underbrace{\min\{\Delta_{t,2,1}\}}_{=0} + \underbrace{\min\{\Delta_{t,1,2}\}}_{=\hat{S}-\hat{T}} = \hat{T} + (\hat{S} - \hat{T}) = \hat{S}, \tag{10}$$

$$\hat{u}_{t,2}^{\mathrm{DRIVE}} = \hat{u}_{t,2} - \underbrace{\min\{\Delta_{t,1,2}\}}_{=\hat{S}-\hat{T}} + \underbrace{\min\{\Delta_{t,2,1}\}}_{=0} = \hat{S} - (\hat{S} - \hat{T}) = \hat{T}. \tag{11}$$

Thus, the unilateral defection step $(\hat{T}, \hat{S})$ is reshaped to $(\hat{S}, \hat{T})$: the defector receives the sucker payoff and the cooperator the temptation payoff. Eqs. (10–11) explicitly demonstrate how the TD gate together with the DRIVE shaping rule realizes the $T \leftrightarrow S$ swap in a single exchange step. By symmetry, the profile $(C, D)$ is reshaped from $(\hat{S}, \hat{T})$ to $(\hat{T}, \hat{S})$.

**Resulting incentives and equilibrium interpretation**  Collecting all action profiles, DRIVE induces the modified payoff matrix with $\hat{T} > \hat{R} > \hat{P} > \hat{S}$. Hence, if the opponent plays $C$, responding with $C$ yields

|   | $C$ | $D$ |
|---|---|---|
| $C$ | $\hat{R}, \hat{R}$ | $\hat{S}, \hat{T}$ |
| $D$ | $\hat{T}, \hat{S}$ | $\hat{P}, \hat{P}$ |

$\hat{R} > \hat{S}$, and if the opponent plays $D$, responding with $C$ yields $\hat{T} > \hat{P}$. Cooperation is therefore a strict best response to both actions, so $(C, C)$ is the unique Nash equilibrium.

**Takeaway**  Under the TD gate (Eq. 3) and shaping rule (Eq. 5), a single request/response exchange in a unilateral defection state deterministically maps $(\hat{T}, \hat{S}) \mapsto (\hat{S}, \hat{T})$, eliminating greed and fear incentives in the transformed PD for that step. This microscopic mechanism directly underpins the equilibrium-level result that DRIVE turns the Prisoner's Dilemma into a coordination game with cooperation as the unique Nash equilibrium, providing a worked-out complement to the proof of Theorem 1.

### A.2 Long-term incentives using average rewards

DRIVE compares a requester's instantaneous reward to the responder's epoch-average reward $\overline{u}_j$ (Eq. 8):

$$\overline{u}_{t,j} = \frac{1}{t+1} \sum_{k=0}^{t} \hat{u}_{k,j}. \tag{12}$$

This choice is not unique: one could instead compare instantaneous rewards on both sides, or use averages on both sides. We analyze the resulting design space and argue that the *average–instant* comparison used in DRIVE is a robust compromise between responsiveness and stability.

**Instant–instant ($\hat{u}_{t,j} - \hat{u}_{t,i}$).** If we replaced $\overline{u}_j$ by $\hat{u}_{t,j}$, then $\Delta_{t,i,j} = \hat{u}_{t,j} - \hat{u}_{t,i}$ would react to single-step noise. A single lucky outcome can generate large transfers, making incentives high-variance and easy to game (e.g., via short-term risk-taking) even when long-run behavior is unchanged.

**Instant–average (compare responder instant to requester average).** This tends to implement *risk sharing*: a responder that just experienced a low instantaneous reward may penalize others even if it is not disadvantaged on average. In social dilemmas, this can blur the distinction between exploitation and variance, creating transfers driven by short-term fluctuations.

**Average–average (compare averages on both sides).** Very stable but slow: it only reacts after the averages of both agents separate substantially. This delays punishment/compensation and can fail to respond quickly to sudden changes in behavior (e.g., a defection episode starting mid-epoch).

**Average–instant (DRIVE).** By comparing the requester's instantaneous reward to the responder's epoch average, DRIVE stays responsive to *current* potentially exploitative outcomes (large request) while using the responder's average to decide whether this outcome reflects a *systematic* imbalance. This balances responsiveness (instant) with robustness (average) and reduces the sensitivity present in instant–instant schemes. Using $\overline{u}_j$ instead of $\hat{u}_{t,j}$ makes DRIVE respond to *systematic* advantage, not single outcomes. Still, $\overline{u}_j$ can deviate from the instantaneous payoff in meaningful ways:

- **Responder doing better than the current outcome ($\overline{u}_j > \hat{u}_{t,j}$).** This happens if $j$ has recently enjoyed higher returns in the epoch. In unilateral defection $(D, C)$, this increases $\Delta_{t,i,j} = \overline{u}_j - \hat{T}$ making it *less negative* than $\hat{S} - \hat{T}$, so the penalty to the defector and the boost to the cooperator become *weaker.*

- **Responder doing worse than the current outcome ($\overline{u}_j < \hat{u}_{t,j}$).** This occurs if $j$ has been systematically disadvantaged earlier in the epoch. Then $\Delta_{t,i,j} = \overline{u}_j - \hat{T}$ becomes *more negative*, strengthening the penalty on the requester (defector) and strengthening the compensation to the responder (cooperator).

A key correction to the informal intuition is that $\Delta$ is *not* guaranteed to be negative. Eq. (8) can be positive whenever the responder's epoch-average exceeds the requester's instantaneous reward, i.e., $\overline{u}_j > \hat{u}_{t,i}$. However, in the *exploitation* regime of a Prisoner's Dilemma (PD), the requester receives the maximal one-step payoff $\hat{T}$. As the epoch-average $\overline{u}_j$ is bounded by the same payoff range as one-step rewards, $\overline{u}_j \leq \hat{T}$, and $\Delta_{t,i,j} < 0$ is guaranteed. Hence, in exploitation, $\Delta$ is robustly negative, which yields DRIVE's intended swap. Outside exploitation, $\Delta$ may have either sign and typically yields small or canceling adjustments in symmetric outcomes.

**Lemma 1 ($\Delta < 0$ in unilateral defection).** *Consider unilateral defection $(D, C)$ in a PD with modified payoffs $\hat{T} > \hat{R} > \hat{P} > \hat{S}$, where agent $i$ defects and receives $\hat{u}_{t,i} = \hat{T}$ while agent $j$ cooperates and receives $\hat{u}_{t,j} = \hat{S}$. Let $\overline{u}_j$ be the epoch average of agent $j$ maintained via the uniform running mean (Eq. 12, Alg. 1). Then $\hat{T} > \overline{u}_j \geq \hat{S}$ and therefore $\Delta_{t,i,j} = \overline{u}_j - \hat{T} \in [\hat{S} - \hat{T}, 0[$. Thus, $\Delta_{t,i,j}$ is strictly negative.*

*Proof.* By (12), $\overline{u}_{t,j}$ is the arithmetic mean of the values $\{\hat{u}_{k,j}\}_{k=0}^{t}$. Since all modified payoffs lie in $[\hat{S}, \hat{T}]$, we have $\hat{S} \leq \hat{u}_{k,j} \leq \hat{T}$ and the averages remain in $[\hat{S}, \hat{T}]$. Since at time $t$ we have $\hat{u}_{t,j} = \hat{S}$, at least one summand is strictly smaller than $\hat{T}$, and the arithmetic mean is strictly smaller than $\hat{T}$, implying $\Delta_{t,i,j} < 0$. $\qquad \square$

Analogously, we can derive the following bounds:

| | | | |
|---|---|---|---|
| **Unilateral Defection:** | $\Delta_{DC} \in [\hat{S} - \hat{T}, 0[$ , | with | $\hat{u}_{t,i} = \hat{T}, \hat{u}_{t,j} = \hat{S}$, and $\hat{T} > \overline{u}_j \geq \hat{S}$ (13) |
| **Mutual Defection:** | $\Delta_{DD} \in ]\hat{S} - \hat{P}, \hat{T} - \hat{P}[$ , | with | $\hat{u}_{t,i} = \hat{P}, \hat{u}_{t,j} = \hat{P}$, and $\hat{T} > \overline{u}_j > \hat{S}$ (14) |
| **Mutual Cooperation:** | $\Delta_{CC} \in ]\hat{S} - \hat{R}, \hat{T} - \hat{R}[$ , | with | $\hat{u}_{t,i} = \hat{R}, \hat{u}_{t,j} = \hat{R}$, and $\hat{T} > \overline{u}_j > \hat{S}$ (15) |
| **Unilateral Cooperation:** | $\Delta_{CD} \in ]0, \hat{T} - \hat{S}]$ , | with | $\hat{u}_{t,i} = \hat{S}, \hat{u}_{t,j} = \hat{T}$, and $\hat{T} \geq \overline{u}_j > \hat{S}$ (16) |

**Dominance conditions for the min-aggregation** In multi-agent settings, DRIVE aggregates responses using a minimum across neighbors. While exploitation responses $\Delta_{DC}$ are always negative (Lemma 1), symmetric outcomes $(C, C)$ and $(D, D)$ can produce deltas that are either positive or negative depending on the responder's epoch-average reward. A natural question is therefore: *Under which conditions is the exploitation response $\Delta_{DC}$ guaranteed to dominate the min-aggregation (i.e., be the most negative response)?*

Since $\Delta_{CD} > 0$ (being defected against) always holds, this case can be ignored for the min operator. Moreover, under calibrated critics, the TD gate rarely activates for the exploited agent receiving the minimal payoff $\hat{S}$ (cf. Theorem 7), making this case structurally unlikely. We therefore compare exploitation responses $(D, C)$ with the lower bounds of symmetric outcomes $(C, C)$ and $(D, D)$. Let the responder's epoch-average under exploitation be rewritten as: $\Delta_{t,i,j} = \overline{u}_j - \hat{u}_{t,i} = \hat{u}_{t,j} + \delta - \hat{u}_{t,i}$. The parameter $\delta$ measures how far the exploited agent's mean reward deviates from steady exploitation. In unilateral defection, for instance, $\overline{u}_j = \hat{S} + \delta$, with $\delta \in [0, \hat{T} - \hat{S})$. Therefore, the resulting exploitation response becomes $\Delta_{DC} = \overline{u}_j - \hat{T} = \hat{S} + \delta - \hat{T}$.

**Proposition 1** (Dominance condition for exploitation responses)**.** *Let $\delta$ denote the deviation of the exploited agent's epoch-average reward from $\hat{u}_{t,j}$ during unilateral defection. Then the exploitation response $\Delta_{DC}$ is guaranteed to dominate the* min*-aggregation against if $\delta < \hat{T} - \hat{R}$.*

*Proof.* We compare $\Delta_{DC}$ with the smallest possible responses. The smallest possible response from $(D, D)$ occurs when the responder's mean equals the lower payoff bound $\hat{S}$: $\Delta_{DD}^{\min} = \hat{S} - \hat{P}$. Analogously, the smallest response from $(C, C)$ occurs at $\Delta_{CC}^{\min} = \hat{S} - \hat{R}$. Thus, exploitation dominates if

$$\Delta_{DC} < \Delta_{DD}^{\min} \iff \hat{S} + \delta - \hat{T} < \hat{S} - \hat{P} \iff \delta < \hat{T} - \hat{P}, \tag{17}$$

$$\Delta_{DC} < \Delta_{CC}^{\min} \iff \hat{S} + \delta - \hat{T} < \hat{S} - \hat{R} \iff \delta < \hat{T} - \hat{R}. \tag{18}$$

$\square$

The derived condition bounds how far the exploited agent's mean reward may deviate from the steady exploitation value $\hat{S}$ while still guaranteeing that exploitation responses dominate the min-aggregation. Several observations follow directly: Since $\hat{T} > \hat{R} > \hat{P} > \hat{S}$, we have $\hat{T} - \hat{R} < \hat{T} - \hat{P}$, so the tighter constraint is $\delta < \hat{T} - \hat{R}$. This threshold corresponds exactly to the *temptation–cooperation gap* of the Prisoner's Dilemma. Thus, as long as the exploited agent's average reward does not improve beyond the cooperation level $\hat{R}$, exploitation responses remain the most negative responses available. In steady exploitation ($\delta \approx 0$), the condition is trivially satisfied and $\Delta_{DC} \approx \hat{S} - \hat{T}$, guaranteeing strong dominance in the min-aggregation. Even if the exploited agent's mean increases due to occasional cooperative outcomes earlier in the epoch, the dominance of exploitation responses persists until the deviation approaches the cooperation payoff scale. Consequently, the intended DRIVE mechanism is robust to moderate within-epoch fluctuations of the mean reward and only fails to dominate when the exploited agent's average reward rises close to cooperative levels. In practice, such large deviations are unlikely in sustained exploitation regimes, meaning the exploitation response typically remains the most negative response and therefore governs the min-aggregation.

### A.3 Implications by the TD gating mechanism

In a PD, defecting against a cooperator yields the maximal one-step payoff $\hat{T}$. If the critic is bounded and roughly calibrated to on-policy returns, it will not systematically predict values exceeding this achievable maximum by more than a small estimation error. Thus, the realized reward is at least as good as expected, making the TD residual non-negative and activating the request gate.

**Theorem 7** (Unilateral defection triggers the TD gate under bounded critics)**.** *Consider a (repeated) Prisoner's Dilemma with modified payoffs $\hat{T} > \hat{R} > \hat{P} > \hat{S}$, i.e., $\hat{u}_{t,i} \leq \hat{T}$ for all joint actions at time $t$. Suppose that at time $t$ the joint action is unilateral defection $(D, C)$, so agent $i$ receives $\hat{u}_{t,i} = \hat{T}$. Assume further that the critic $\hat{V}_i$ is bounded and not over-optimistic at the current history in the sense that there exists $\epsilon \geq 0$ with $\hat{V}_i(\tau_{t,i}) \leq \hat{T} - \epsilon$, and that the continuation value is bounded below by some $V_{\min}$, i.e., $\hat{V}_i(\tau) \geq V_{\min}$ for all histories $\tau$. Then the TD residual is non-negative whenever $\epsilon \geq -\gamma V_{\min}$, and the TD gate activates.*

*Proof.* Under unilateral defection $(D, C)$, agent $i$ obtains the maximal one-step reward $\hat{u}_{t,i} = \hat{T}$. Using the TD residual according to Eq. (3) with the above assumptions on the critic being bounded and not over-optimistic, yields

$$TD_i(\hat{u}_{t,i}) \geq \hat{T} + \gamma V_{\min} - (\hat{T} - \epsilon) = \epsilon + \gamma V_{\min}.$$

Therefore, if $\epsilon \geq -\gamma V_{\min}$ (in particular if $V_{\min} \geq 0$ or $\epsilon$ is chosen to dominate the worst-case negative continuation term), then $TD_i(\hat{u}_{t,i}) \geq 0$ and the DRIVE request gate activates. $\square$

Under return normalization, rewards and value targets are centered and scaled within each epoch, keeping both $\hat{u}_{t,i}$ and $\hat{V}_i(\tau_{t,i})$ in a controlled numerical range. Actor–critic training further encourages $\hat{V}_i$ to track realized returns rather than extrapolate beyond attainable outcomes. Hence, boundedness and the absence of persistent over-optimism are natural regularity conditions.

**Over-activation of the TD gate and its shaping effect.** While the previous result addresses *under-activation* (failing to request when exploiting), the TD gate may also activate in situations where $TD_i(\hat{u}_{t,i}) \geq 0$ without exploitation. To illustrate the most extreme shaping effect, assume a two-agent scenario (i.e., no min aggregation), where the TD gates *always* open, i.e., both agents request whenever possible, which yields:

$$u_{t,i}^{\text{DRIVE}} = \hat{u}_{t,i} - \Delta_{t,j,i} + \Delta_{t,i,j} = \hat{u}_{t,i} - (\overline{u}_i - \hat{u}_{t,j}) + (\overline{u}_j - \hat{u}_{t,i}) = \hat{u}_{t,j} + \overline{u}_j - \overline{u}_i. \tag{19}$$

Thus, in the worst case, the instantaneous rewards are *swapped* and additionally shifted by the difference of epoch averages. We now bound the shaped reward ranges under this failure mode.

In a PD with one-step payoffs $\hat{T} > \hat{R} > \hat{P} > \hat{S}$, any epoch average satisfies $\hat{S} \leq \overline{u}_i \leq \hat{T}$ for all agents $i$, since $\overline{u}_i$ is an average of realized one-step rewards in $\{\hat{S}, \hat{P}, \hat{R}, \hat{T}\}$ (or more generally in $[\hat{S}, \hat{T}]$). For additional simplification, we denote the payoff range as $\Gamma := \hat{T} - \hat{S}$. Let $(\hat{u}_{t,i}, \hat{u}_{t,j})$ be the instantaneous modified rewards under the current action profile. Using (19), we can analyze the worst-case DRIVE reward bounds in all cases assuming no gating:

- **Mutual cooperation** $(\hat{R}, \hat{R})$: $u_{t,i}^{\text{DRIVE}} = \hat{R} + \overline{u}_j - \overline{u}_i \in \,]\hat{R} - \Gamma, \ \hat{R} + \Gamma[$. Particularly, if both agents have similar averages (typical in a cooperative regime, $\overline{u}_i \approx \overline{u}_j \approx \hat{R}$), then $u_{t,i}^{\text{DRIVE}} \approx \hat{R}$.

- **Mutual defection** $(\hat{P}, \hat{P})$: $u_{t,i}^{\text{DRIVE}} = \hat{P} + \overline{u}_j - \overline{u}_i \in \,]\hat{P} - \Gamma, \ \hat{P} + \Gamma[$. Again, if $\overline{u}_i \approx \overline{u}_j \approx \hat{P}$, then $u_{t,i}^{\text{DRIVE}} \approx \hat{P}$.

- **Unilateral defection** $(\hat{T}, \hat{S})$ ($i$ **defects**): Using $\overline{u}_i \in [\hat{T}, \hat{S}[$ and $\overline{u}_j \in ]\hat{T}, \hat{S}]$ ungated shaping yields:

$$2\hat{S} - \hat{T} = \hat{S} - (\hat{T} - \hat{S}) \ \leq \ u_{t,i}^{\text{DRIVE}} \ < \ \hat{S} + (\hat{T} - \hat{S}) = \hat{T}.$$

  Thus, even if both gates erroneously activate, the defector's shaped reward remains strictly below $\hat{T}$. In steady state ($\overline{u}_i \approx \hat{T}$, $\overline{u}_j \approx \hat{S}$), the mechanism intensifies the intended swap, $u_{t,i}^{\text{DRIVE}} \approx 2\hat{S} - \hat{T} < \hat{S}$, thereby strengthening the penalty for exploitation. If $\overline{u}_i$ decreases because agent $i$ cooperated earlier in the epoch, the penalty is softened but remains bounded above by $\hat{T}$. Hence, even without gating and under mean deviations, defection is never rewarded.

- **Being defected against** $(\hat{S}, \hat{T})$ ($i$ **cooperates**): Using the bounds $\overline{u}_i \in [\hat{S}, \hat{T}[$ and $\overline{u}_j \in ]\hat{S}, \hat{T}]$, we obtain

$$\hat{S} \ < \ u_{t,i}^{\text{DRIVE}} \ \leq \ 2\hat{T} - \hat{S}.$$

  Thus, in the ungated scenario, the exploited cooperator's reward can exceed $\hat{T}$ (up to $2\hat{T} - \hat{S}$). In steady unilateral defection ($\overline{u}_i \approx \hat{S}$, $\overline{u}_j \approx \hat{T}$), this corresponds to the intensified swap $u_{t,i}^{\text{DRIVE}} \approx 2\hat{T} - \hat{S} > \hat{T}$. Hence, over-activation may overcompensate the cooperator. This illustrates why TD gating (and, in the multi-agent case, min aggregation) is structurally important: it suppresses such inflation and restricts shaping to genuine exploitation signals.

This analysis isolates the *extreme* scenario in which the TD gate fails completely, and exchange always occurs in both directions. In practice, the TD gate and (in multi-agent settings) the min aggregation reduce the likelihood that large positive transfers dominate, because they suppress exchanges when the requester's gate is closed and preferentially select the most negative (i.e., most compensatory) response among neighbors.

**Summary.** Over-activation can occur due to critic noise, temporary miscalibration, or reward shifts. However, because DRIVE shapes rewards via *differences* relative to peer averages and aggregates via min, symmetric or near-symmetric outcomes produce negligible deltas. As a result, unintended gate openings have limited structural impact. Persistent large distortions would indicate critic instability rather than a flaw of the gating mechanism itself.

## A.4 Detailed Proof of Theorem 1

*Proof.* Consider a two-agent PD with $\hat{T} > \hat{R} > \hat{P} > \hat{S}$. Under unilateral defection $(D, C)$, agent $i$ (defector) receives $\hat{u}_{t,i} = \hat{T}$ and agent $j$ (cooperator) receives $\hat{u}_{t,j} = \hat{S}$.

1. **Gated interaction.** By Theorem 7, under bounded and rational critics, the TD gate activates for the exploiting agent $i$ and does not activate for $j$. Hence, only $i$ issues a request and we obtain

$$\Delta_{t,i,j} = \overline{u}_j - \hat{T}, \qquad \Delta_{t,j,i} = 0.$$

2. **Sign and bounds of the response.** By Lemma 1, $\overline{u}_j \in [\hat{S}, \hat{T}[$, hence the response is strictly negative.

3. **Resulting shaped rewards.** Using Eq. (5) (two-agent case),

$$u_{t,i}^{\text{DRIVE}} = \hat{T} + \Delta_{t,i,j} = \overline{u}_j, \qquad u_{t,j}^{\text{DRIVE}} = \hat{S} - \Delta_{t,i,j} = \hat{T} - (\overline{u}_j - \hat{S}).$$

   Thus, DRIVE does not deterministically map $(\hat{T}, \hat{S}) \mapsto (\hat{S}, \hat{T})$, but instead yields the transformed payoffs $(\overline{u}_j, \ \hat{T} - (\overline{u}_j - \hat{S}))$.

4. **Sufficient condition for incentive reversal.** For cooperation to be a best response against a cooperator, we require $u_{t,i}^{\text{DRIVE}} = \overline{u}_j < \hat{R}$. Under this condition, defecting yields strictly less payoff than mutual cooperation $(\hat{R}, \hat{R})$, eliminating the temptation to defect.

5. **Realization under typical dynamics.** Appendix A.2 shows that in sustained unilateral defection, $\overline{u}_j \approx \hat{S}$, so the condition $\overline{u}_j < \hat{R}$ holds strictly and $u_{t,i}^{\text{DRIVE}} \approx \hat{S}$, $u_{t,j}^{\text{DRIVE}} \approx \hat{T}$, recovering the payoff inversion.

6. **Symmetric outcomes.** For $(C, C)$ and $(D, D)$, Appendix A.2 shows that $\overline{u}_i \approx \overline{u}_j$, yielding $\Delta \approx 0$ and preserving $(\hat{R}, \hat{R})$ and $(\hat{P}, \hat{P})$.

Combining these results, DRIVE removes both greed $(\hat{R} > \overline{u}_j)$ and fear $(\hat{T} > \hat{P})$, making mutual cooperation the dominant strategy under DRIVE. $\square$

## A.5 Analysis of Protocol Compliance Cases

To complement the discussion in Sec. 5, we present a worked example for a two-player Prisoner's Dilemma under DRIVE with different levels of protocol compliance. We consider generic PD payoffs $T > R > P > S$ as in Eq. (1); for concreteness, one may think of the canonical values $(T, R, P, S) = (5, 3, 1, 0)$, but none of the arguments below depend on these particular numbers. In each case, agent $i$ defects while agent $j$ cooperates, and DRIVE applies reward shaping based on request/response behavior.

| Case | Defector payoff | Cooperator payoff |
|---|---|---|
| (1) Full compliance | $S$ | $T$ |
| (2) Defector does not send request | $T$ (no penalty) | $S$ |
| (3) Cooperator withholds response | $T$ (no penalty) | $S$ |
| (4) Misreported request / response | may collapse penalties | may lead to mis-penalization |
| (5) No compliance at all | $T$ | $S$ |

Table 1: Illustration of payoffs under different protocol compliance scenarios in a 2-player PD.

**(1) Full compliance** If all agents truthfully follow the protocol, the results of Theorem 1 apply. Unilateral defection $(T, S)$ is reshaped into $(S, T)$: the defector receives $S$ and the cooperator $T$, eliminating the incentive to defect.

**(2) Defector does not send a request** If the defector refrains from sending a request despite a positive TD, it avoids penalization from neighbors but also loses the chance of mutual improvement. Its payoff remains $T$, while the cooperator's payoff remains $S$. This strategy indirectly penalizes the non-requesting agent, which cannot gain from reciprocal shaping.

**(3) Cooperator withholds a response**  If the cooperator fails to respond (intentionally or due to loss), the defector avoids the intended penalty. The payoffs revert to $(T, S)$. Because of the min-aggregation across neighbors, the absence of a single response only matters if it is the unique or strongest penalty.

**(4) Misreporting**  Misreporting can occur in two forms: (i) agents send requests despite having no improvement signal (TD< 0), or (ii) agents respond with incorrect values of $\Delta$ (due to error or intent). In the first case, the resulting differences are typically small or positive, which the min operator filters out, making the effect negligible. In the second case, however, false responses can distort the aggregation: A misreporting neighbor may reduce or nullify the penalty for a defector, or unjustly penalize a compliant requester. While this can destabilize cooperation in the worst case, it is not worse than the vulnerabilities faced by other PI methods. Overall, misreporting represents a robustness limitation but does not fundamentally break the mechanism as long as the majority of responses remain truthful.

**(5) No compliance at all**  If no requests or responses are exchanged, all shaping terms vanish. DRIVE collapses to plain MARL with payoffs $(T, S)$ under unilateral defection, and cooperation incentives are lost.

**Extension to larger scenarios**  The two-player example above presents the limiting case in which a single non-compliant agent corresponds to 50% protocol adherence, thereby strongly magnifying the impact of individual deviations. In larger populations, however, the effect of partial non-compliance is more gradual. As long as a sufficient fraction of agents continue to respond truthfully, the min-aggregation still enforces penalization of defectors.

**Three-player example**  Consider a group of three agents, where agent $i$ defects and agents $j$ and $k$ cooperate. Each cooperator truthfully responds with $\Delta_{i,j}, \Delta_{i,k} < 0$. The defector's shaped reward is $u_i^{\text{DRIVE}} = T + \min\{\Delta_{i,j}, \Delta_{i,k}\}$, so as long as at least one cooperator responds truthfully, $u_i^{\text{DRIVE}}$ is penalized. If one cooperator withholds, the other still enforces the penalty, though it may be weaker. Thus, unlike in the 2-player case, cooperation incentives do not collapse entirely when a single agent is non-compliant.

## A.6  Continuous mean-field convergence analysis

To complement the discrete profile analysis of DRIVE in the 2-player PD, we derive a continuous mean-field view in terms of cooperation probabilities. This makes the previous implicit convergence intuition explicit: Under calibrated critics, DRIVE induces a smooth payoff field that points toward mutual cooperation.

**Setting.**  Consider a 2-player PD with payoffs $T > R > P > S$. Let agent $i$ cooperate with probability $\theta_i \in [0, 1]$ and agent $j$ with probability $\theta_j \in [0, 1]$, independently across rounds within an epoch. The action-profile probabilities are: $p_{CC} = \theta_i \theta_j,$ $p_{CD} = \theta_i(1 - \theta_j),$ $p_{DC} = (1 - \theta_i)\theta_j,$ $p_{DD} = (1 - \theta_i)(1 - \theta_j)$.

**Lemma 2** (Expected one-step rewards $\mu$).  *Under mixed cooperation probabilities* $(\theta_i, \theta_j)$ $\mu_i$ *and* $\mu_j$ *are:*

$$\mu_i(\theta_i, \theta_j) = \theta_i \theta_j R + \theta_i(1 - \theta_j)S + (1 - \theta_i)\theta_j T + (1 - \theta_i)(1 - \theta_j)P, \tag{20}$$

$$\mu_j(\theta_i, \theta_j) = \theta_i \theta_j R + \theta_i(1 - \theta_j)T + (1 - \theta_i)\theta_j S + (1 - \theta_i)(1 - \theta_j)P. \tag{21}$$

*Proof.* Immediate by conditioning on the four joint action profiles and weighting by their probabilities.  □

**Assumption 1** (Epoch averages).  *Let an epoch contain* $H$ *independent repetitions of the mixed profile* $(\theta_i, \theta_j)$ *and let*

$$\overline{u}_i = \frac{1}{H}\sum_{h=1}^{H} u_i^{(h)}, \qquad \overline{u}_j = \frac{1}{H}\sum_{h=1}^{H} u_j^{(h)}.$$

*Then* $\mathbb{E}[\overline{u}_i \mid \theta_i, \theta_j] = \mu_i(\theta_i, \theta_j)$ *and* $\mathbb{E}[\overline{u}_j \mid \theta_i, \theta_j] = \mu_j(\theta_i, \theta_j)$. *For moderate or large* $H$, *the running means concentrate around these expectations by the law of large numbers, so the deterministic mean-field approximation becomes accurate.*

First, we show that increasing one's own cooperation probability is never locally beneficial in the base game.

**Lemma 3** (Unshaped incentives favor defection). *Define $\kappa(\theta) := \theta(T-R) + (1-\theta)(P-S)$. Then $\kappa(\theta) > 0$ for all $\theta \in [0,1]$, and*

$$\frac{\partial \mu_i}{\partial \theta_i}(\theta_i, \theta_j) = -\kappa(\theta_j) < 0, \qquad \frac{\partial \mu_j}{\partial \theta_j}(\theta_i, \theta_j) = -\kappa(\theta_i) < 0, \qquad on\ [0,1]^2. \tag{22}$$

*Proof.* Differentiating $\mu_i$ (Eq. 20) with respect to $\theta_i$ yields $\frac{\partial \mu_i}{\partial \theta_i} = \theta_j R + (1-\theta_j)S - \theta_j T - (1-\theta_j)P = \theta_j(R-T) + (1-\theta_j)(S-P) = -\kappa(\theta_j)$. Since $T > R$ and $P > S$, we have $\kappa(\theta) > 0$ on $[0,1]$. The result for agent $j$ is symmetric. $\qquad\square$

**Intuition.** The quantity $\kappa(\theta_j)$ is the expected loss incurred by agent $i$ when increasing its cooperation probability while holding the opponent's behavior fixed. It is a convex combination of the two temptation gaps $T - R$ and $P - S$, corresponding to the cases where the opponent cooperates or defects, respectively. Since both gaps are strictly positive in the PD, increasing cooperation is always locally disadvantageous in the unshaped game.

**Gated mean-field DRIVE.** In the 2-agent case, the DRIVE reward of agent $i$ can be written as

$$u_{t,i}^{\text{DRIVE}} = \hat{u}_{t,i} + \Delta_{t,i,j} - \Delta_{t,j,i},$$

where, by Eq. (8), each term is present only if the corresponding request gate is active. At the microscopic level, this depends on the realized TD residuals and running means. To obtain a continuous approximation, we summarize the *expected gating contribution* at the level of $(\theta_i, \theta_j)$. Using Assumption 1

$$\mathbb{E}[\Delta_{i,j} \mid \theta_i, \theta_j] = \mathbb{E}[\overline{u}_j - u_i \mid \theta_i, \theta_j] = \mu_j(\theta_i, \theta_j) - \mu_i(\theta_i, \theta_j) = -\mathbb{E}[\overline{u}_i - u_j \mid \theta_i, \theta_j] = -\mathbb{E}[\Delta_{i,j} \mid \theta_i, \theta_j].$$

**Assumption 2** (Complementary mean-field gating). *Assume the critics are calibrated in the sense of Theorem 7 and the accompanying discussion: exploitative asymmetries tend to activate the advantaged agent's gate, while the disadvantaged agent's gate remains inactive or weakly active, and symmetric regimes induce weak or cancelling gate activity. At the mean-field level, the gating contribution equals the reward asymmetry:*

$$\mathbb{E}[\mathbf{1}\{\text{TD}_i \geq 0\}\Delta_{i,j} - \mathbf{1}\{\text{TD}_j \geq 0\}\Delta_{j,i} \mid \theta_i, \theta_j] = \mu_j(\theta_i, \theta_j) - \mu_i(\theta_i, \theta_j).$$

**Interpretation.** Assumption 2 is a parameter-free mean-field closure of the request/response mechanism in Eq. (7). It reflects the same qualitative behavior established earlier in the appendix: under unilateral exploitation, the exploiting agent requests while the exploited agent typically does not; under symmetric outcomes, gate activity is weak or cancelling. Thus, in expectation, the net crossed-gating contribution behaves like a single effective correction term equal to the current reward asymmetry.

**Lemma 4** (Gated mean-field DRIVE rewards). *Under Assumption 2, the expected shaped rewards are*

$$\widetilde{\mu}_i^{\text{G}}(\theta_i, \theta_j) = \mu_i(\theta_i, \theta_j) + \big(\mu_j(\theta_i, \theta_j) - \mu_i(\theta_i, \theta_j)\big) = \mu_j(\theta_i, \theta_j), \tag{23}$$

$$\widetilde{\mu}_j^{\text{G}}(\theta_i, \theta_j) = \mu_j(\theta_i, \theta_j) + \big(\mu_i(\theta_i, \theta_j) - \mu_j(\theta_i, \theta_j)\big) = \mu_i(\theta_i, \theta_j). \tag{24}$$

*Proof.* Immediate from Assumption 2. $\qquad\square$

**Lemma 5** (Gradient of the gated mean-field reward). *Define $\lambda(\theta) := \theta(R-S) + (1-\theta)(T-P)$. Then $\lambda(\theta) > 0$ for all $\theta \in [0,1]$, and*

$$\frac{\partial \widetilde{\mu}_i^{\text{G}}}{\partial \theta_i}(\theta_i, \theta_j) = \lambda(\theta_j) > 0, \qquad \frac{\partial \widetilde{\mu}_j^{\text{G}}}{\partial \theta_j}(\theta_i, \theta_j) = \lambda(\theta_i) > 0. \tag{25}$$

*Proof.* By Lemma 4, $\widetilde{\mu}_i^{\text{G}}(\theta_i, \theta_j) = \mu_j(\theta_i, \theta_j)$. Differentiating with respect to $\theta_i$ gives $\frac{\partial \widetilde{\mu}_i^{\text{G}}}{\partial \theta_i} = \frac{\partial \mu_j}{\partial \theta_i}$. Using the definition of $\mu_j$ from Lemma 2, $\mu_j(\theta_i, \theta_j) = \theta_i \theta_j R + \theta_i(1-\theta_j)T + (1-\theta_i)\theta_j S + (1-\theta_i)(1-\theta_j)P$, so $\frac{\partial \mu_j}{\partial \theta_i} = \theta_j(R-S) + (1-\theta_j)(T-P) = \lambda(\theta_j)$. Since $R > S$ and $T > P$, we have $\lambda(\theta) > 0$ for all $\theta \in [0,1]$. The expression for $j$ is symmetric. $\qquad\square$

**Theorem 8** (Global convergence under complementary mean-field gating)**.** *Under Assumption 2, every interior trajectory of the following dynamics converges to* $(1,1)$*:*

$$\dot{\theta}_i = \theta_i(1 - \theta_i)\frac{\partial \widetilde{\mu}_i^{\mathrm{G}}}{\partial \theta_i}, \qquad \dot{\theta}_j = \theta_j(1 - \theta_j)\frac{\partial \widetilde{\mu}_j^{\mathrm{G}}}{\partial \theta_j}, \tag{26}$$

*Proof.* By Lemma 5, $\frac{\partial \widetilde{\mu}_i^{\mathrm{G}}}{\partial \theta_i} > 0$ and $\frac{\partial \widetilde{\mu}_j^{\mathrm{G}}}{\partial \theta_j} > 0$ for all $(\theta_i, \theta_j) \in [0,1]^2$. Hence in the open unit square, $\dot{\theta}_i > 0$ and $\dot{\theta}_j > 0$. Thus both coordinates are monotone increasing and bounded above by 1, so both converge. Any limit point with $\theta_i < 1$ or $\theta_j < 1$ would still have strictly positive drift, which is impossible. Therefore the only possible limit is $(1,1)$. □

**Relation to the payoff-swap analysis.** This mean-field result is consistent with the microscopic payoff-swap mechanism derived in Appendix A.1. At the discrete unilateral-defection extremes $(D, C)$ and $(C, D)$, DRIVE swaps the payoffs $T$ and $S$ in a single exchange. The present mean-field closure extends this idea from the four discrete corners to the full cooperation-probability square: in expectation, each agent optimizes the other agent's reward, and the resulting gradient field points monotonically toward mutual cooperation.

**Alternative interpolations.** Assumption 2 yields the simplest parameter-free interpolation. One may also consider richer mean-field kernels of the form $\widetilde{\mu}_i(\theta_i, \theta_j) = \mu_i(\theta_i, \theta_j) + \beta(d)\big(\mu_j(\theta_i, \theta_j) - \mu_i(\theta_i, \theta_j)\big)$, with $d = |\theta_i - \theta_j|$ to model stronger correction as asymmetry grows. For instance, $\beta(d) = 1 + d$ preserves the same endpoint behavior while inducing a nonlinear interpolation. Such variants may be useful for visualization or refined modeling, but Assumption 2 provides the cleanest closed-form approximation and requires no additional free parameters.

While variants may be useful for refined modeling, note that Assumption 2 models the constant choice $\beta(d) \equiv 1$, which corresponds to one effective gate opening per pair in expectation.

**Finite-grid no-regret corollary.** To connect the smooth vector field to a regret-style guarantee, discretize the cooperation interval as $\Theta_h := \{0, h, 2h, \dots, 1\}$, with $h = 1/K$ for some integer $K \geq 1$, and consider the induced finite game with utilities

$$(\theta_i, \theta_j) \mapsto \widetilde{\mu}_i^G(\theta_i, \theta_j), \qquad (\theta_i, \theta_j) \mapsto \widetilde{\mu}_j^G(\theta_i, \theta_j).$$

**Proposition 2** (No-regret concentration on cooperation)**.** *Under the condition of Theorem 8, the discretized game on* $\Theta_h \times \Theta_h$ *has the unique Nash equilibrium* $(1,1)$*. Moreover,* $(1,1)$ *is the unique coarse correlated equilibrium. Therefore, any Hannan-consistent no-regret learning dynamics on this discretized game have empirical play that converges to* $(1,1)$*.*

*Proof.* By Theorem 8, the utility of each player $\widetilde{\mu}_i^{\mathrm{G}}(\cdot, \theta_j)$ is strictly increasing in its own cooperation probability $\theta_i$ for any fixed opponent action $\theta_j$. Hence the unique best response of each player is $\theta = 1$, so $(1,1)$ is the unique Nash equilibrium.

To show uniqueness of the *coarse correlated equilibrium* (CCE), let $\nu$ be any distribution over $\Theta_h \times \Theta_h$ that assigns positive mass to a profile with $\theta_i < 1$. Because $\widetilde{\mu}_i^{\mathrm{G}}(\cdot, \theta_j)$ is strictly increasing for every fixed $\theta_j$, player $i$ can improve its expected utility by deviating to action 1, contradicting the CCE condition. Thus any CCE must assign probability one to $\theta_i = 1$. Applying the same argument to player $j$ shows that any CCE must assign probability one to $(1,1)$. Hence $(1,1)$ is the unique CCE. Standard no-regret results for finite games imply convergence of empirical play to the set of CCEs, which here is the singleton $\{(1,1)\}$. □

**What this does and does not prove.** The above analysis establishes a continuous mean-field convergence result and a corresponding finite-grid no-regret guarantee under complementary mean-field gating under calibrated critics. It does *not* yet prove convergence of the full actor–critic implementation. To close that gap, one would additionally need to show that (i) critic alignment is maintained throughout training, (ii) the running means concentrate around $\mu_i, \mu_j$ sufficiently fast, and (iii) the stochastic actor updates track the limiting vector field above.

## A.7 Extension to a Simple N-Agent PD Setting

The main theorem focuses on the 2-agent Prisoner's Dilemma (PD). Here we first provide a clean extension to an $N$-agent *graphical* PD under full protocol compliance, where payoffs decompose into symmetric 2-player PD interactions, and then progressively relax structural and behavioral assumptions.

**Definition 3** (Graphical PD). Let $G = (V, E)$ be an undirected graph with $|V| = N$ agents. Each agent $i \in V$ chooses an action $a_i \in \{C, D\}$. For every edge $(i, j) \in E$ we associate a 2-player PD with payoffs $(R, R)$, $(P, P)$, $(T, S)$, $(S, T)$ as in Eq. (1). The stage payoff of agent $i$ is $u_i(a) = \sum_{j \in N(i)} u_{i,j}(a_i, a_j)$, where $N(i)$ is the neighborhood of $i$ in $G$ and $u_{i,j}$ is the PD payoff against neighbor $j$.

To isolate the effect of incentive alignment, we assume that each pairwise interaction is shaped independently using the 2-agent DRIVE protocol and that all agents comply fully with the request and response rules. That is, in every interaction with neighbor $j$, agent $i$ behaves exactly as in the 2-agent analysis of Appendix A.1.

**Proposition 3** (Pairwise incentive alignment). *Consider a graphical PD as above and assume that each edge $(i, j)$ uses the 2-agent DRIVE protocol with shaping rule (7) applied independently to $u_{i,j}(a_i, a_j)$. Then, for every agent $i$ and every joint action profile $a_{-i}$ of its neighbors,*

$$u_i^{\mathrm{DRIVE}}(C, a_{-i}) > u_i^{\mathrm{DRIVE}}(D, a_{-i}),$$

*so cooperation is a dominant action for every agent and the unique Nash equilibrium of the shaped game is the all-cooperate profile $(C, \ldots, C)$.*

*Proof.* By Theorem 1 and its detailed proof in Appendix A.1, for each neighbor $j$ the pairwise DRIVE-shaped payoff function $u_{i,j}^{\mathrm{DRIVE}}$ makes $C$ strictly dominate $D$ in the corresponding 2-player PD. That is, for any fixed $a_j \in \{C, D\}$, $u_{i,j}^{\mathrm{DRIVE}}(C, a_j) > u_{i,j}^{\mathrm{DRIVE}}(D, a_j)$. Summing these strict inequalities over all neighbors $j \in N(i)$ for any joint neighbor action profile $a_{-i}$ yields

$$u_i^{\mathrm{DRIVE}}(C, a_{-i}) = \sum_{j \in N(i)} u_{i,j}^{\mathrm{DRIVE}}(C, a_j) > \sum_{j \in N(i)} u_{i,j}^{\mathrm{DRIVE}}(D, a_j) = u_i^{\mathrm{DRIVE}}(D, a_{-i}).$$

Hence $C$ is a strict best response to any $a_{-i}$ for all agents $i$, so $(C, \ldots, C)$ is the unique Nash equilibrium. $\square$

This proposition formalizes how the 2-player incentive alignment result extends to a simple class of $N$-agent games with additive pairwise PD interactions. Our SSD experiments instantiate more complex interaction patterns, but the graphical PD result provides a clean theoretical illustration of how DRIVE scales beyond two agents.

**Pairwise PD interactions across neighborhoods** In many multi-agent systems, agents repeatedly engage in pairwise social dilemmas with dynamically changing or randomly matched partners (an *interaction graph*), while incentive exchanges are restricted to a possibly different and fixed *communication graph* defined by neighborhoods $N_{t,i}$. In such settings, the DRIVE-shaped reward of a defector $i$ still takes the form $u_i^{\mathrm{DRIVE}} = T + \min_{j \in N_{t,i}} \Delta_{i,j}$. Whenever the exploited cooperator in a given PD round is also a neighbor of $i$, the exact $T \leftrightarrow S$ payoff swap from the 2-agent case is recovered. When interaction partners lie outside $N_{t,i}$, the penalty is instead delivered by whichever compliant neighbors suffer a sustained long-run disadvantage from $i$'s behavior (e.g., through shared resources or future interactions). Thus, the interaction graph determines *which behaviors create negative externalities*, while the communication graph determines *who can hold whom accountable*. As long as each defector has at least one compliant neighbor whose long-run return is reduced when it defects, the sign of the incentive remains cooperative, even if the magnitude differs from the fully local 2-agent PD.

**Beyond graphical PDs** Many benchmark social dilemma environments, such as Coin Game or Harvest, do not admit an explicit decomposition into pairwise PD payoffs. Instead, incentives arise indirectly through shared resources, delayed consequences, and aggregate system dynamics. Nevertheless, the same principle applies: agents whose behavior systematically reduces their neighbors' long-run returns generate negative differences $\Delta_{i,j}$ and are penalized by DRIVE. This perspective explains why DRIVE empirically promotes cooperation in such environments, even in the absence of an explicit PD structure.

**General $N$-agent compliance and robustness**   The preceding results assume full compliance with the protocol. We now consider a general $N$-agent setting in which only $M$ agents truthfully follow the DRIVE protocol, while $K = N - M$ agents may behave adversarially by withholding requests or responses or by misreporting. For any game-theoretic defector $i$, effective penalization requires two conditions: (i) $i$ itself must comply with the request gate (sending a request when TD $\geq 0$), and (ii) at least one neighbor $j \in N_i$ must respond truthfully. Under these conditions, the shaped reward of $i$ is

$$u_i^{\mathrm{DRIVE}} = T + \min\{\Delta_{i,j}\}_{j \in N_i, \, j \text{ compliant}}.$$

As long as a compliant responder exists, defection is penalized; the severity depends on the most negative difference among the responders, but the penalty does not collapse entirely. In fully connected populations, this reduces to the simple requirement that at least two agents comply (the requester and one responder), i.e. $M/N \geq 2/N$. In general communication topologies, the compliant set must form a *dominating set*, ensuring that each requester has at least one compliant neighbor. Thus, DRIVE exhibits a *graceful degradation*: as compliance $M/N$ decreases, cooperation incentives weaken smoothly but only collapse completely if no responder is available.

**Connectivity requirements**   In fully connected graphs, penalization is ensured as soon as at least two agents comply (the requester and one responder), corresponding to a compliance rate $c \geq 2/N$. In general communication topologies, the compliant agents must form a *dominating set*: every requester must have at least one compliant neighbor. Formally, for graph $G = (V, E)$ with compliant set $C \subseteq V$, penalization holds iff $\forall i \in V : N(i) \cap C \neq \varnothing$. The minimal compliant set size is the domination number $\gamma(G)$, yielding a topology-aware threshold $|C| \geq \gamma(G)$.

**Adversarial robustness**   The above threshold can be interpreted directly in terms of $K$ adversarial agents. If $K$ agents are adversarial (never providing truthful responses), DRIVE continues to penalize every *protocol-compliant* defector as long as the remaining $M = N - K$ compliant agents contain a dominating set. Equivalently, the system tolerates up to $K \leq N - \gamma(G)$ adversarial agents. However, agents that are adversarial both in their *actions* (always defecting) and their *protocol behavior* (never sending requests) cannot be punished by any PI scheme based on self-reported signals, including DRIVE. To compliant neighbors, these agents appear as players whose defections generate no negative differences; this reduces local cooperative pressure and gradually pushes the system toward the underlying non-PI MARL dynamics (Case 5), but does not cause an abrupt collapse.

Overall, these results highlight DRIVE's property of *graceful degradation*: full compliance yields strong incentive-alignment guarantees, while partial compliance weakens but does not immediately destroy cooperative incentives, provided the communication topology satisfies minimal coverage conditions. In two-agent systems, a single non-compliant agent already reduces compliance to 50%, eliminating penalization. In larger populations ($N > 2$), even if several agents defect from the protocol, the remaining majority can still enforce penalization through the min-aggregation. Hence, DRIVE remains effective as long as the compliance rate stays above the topology-specific threshold, often close to a simple majority in practice. Nevertheless, systematic misreporting or universal non-compliance collapses DRIVE to baseline MARL, emphasizing the importance of designing variants that are robust to missing or adversarial responses in future work.

## A.8   Affine Reward Transformations: Scope and Edge Cases

Here, we expand on the affine reward-change model used in our analysis. In each epoch $m$, the environmental reward is transformed by a shared positive affine map $r' = c_m r + b_m$, applied uniformly to all agents and timesteps. The schedules $c_m$ and $b_m$ may vary arbitrarily across epochs, covering all reward dynamics used in Sec. 8 (linear, exponential, stepwise, and damped–cosine). For illustration, in the damped–cosine setting, we use $c_m = (1 - m/E) \cos^2(2\eta m)$ and $b_m = \eta$, so the effective reward scale oscillates while decaying over time.

Isolated epochs may satisfy $c_m \approx 0$, yielding $r' \approx b_m$ and temporarily collapsing the Prisoner's Dilemma inequalities ($T, R, P, S$ become indistinguishable). The stage game becomes nearly payoff-indifferent in these epochs.

**Transformations not covered** This affine class excludes genuinely non-linear reward changes such as clipping, saturation, sign-dependent remapping, or state-dependent transformations that cannot be written as a shared affine map. These represent a broader family of sim-to-real shifts, which we leave for future work.

## A.9 Reward Changes and Invariance in the PD

We first formalize how affine reward transformations affect the PD inequalities.

**Lemma 6** (Affine transformations preserve PD structure). *Let $T > R > P > S$ and define $T' = cT + b$, $R' = cR + b$, $P' = cP + b$, $S' = cS + b$ for some $c > 0$, $b \in \mathbb{R}$. Then $T' > R' > P' > S'$.*

*Proof.* Since $c > 0$, we have $T' - R' = c(T - R) > 0$, $R' - P' = c(R - P) > 0$, and $P' - S' = c(P - S) > 0$, so $T' > R' > P' > S'$. The additive constant $b$ cancels in all pairwise differences. □

We can now restate Theorems 2 and 3 more formally.

*Proof of Theorem 2.* In MATE, mutual cooperation is individually rational iff the global token $x > 0$ satisfies $x \geq \max\{P - S, (T - R)/3\}$ (Phan et al., 2024). Under the affine transformation of Definition 1, these thresholds become $P' - S'$ and $(T' - R')/3$, i.e.,

$$x \geq \max\{P' - S', (T' - R')/3\}.$$

If we fix any $x$ satisfying this condition, because $c_m$ and $b_m$ are arbitrary (subject to $c_m > 0$), we can always choose $(c_m, b_m)$ such that $P' - S' > x$, e.g., by taking $c_m$ large enough. Then the transformed threshold exceeds $x$, and the MATE payoff matrix no longer guarantees cooperation as a best response. Hence, MATE is not invariant to such reward changes. □

*Proof of Theorem 3.* Let $(R, P, T, S)$ and $(R', P', T', S')$ be two sets of PD payoffs related by any $f_{mod}$ of Definition 1. By Lemma 6, $T > R > P > S$ implies $T' > R' > P' > S'$. The detailed analysis in Appendix A.1 shows that, under full compliance, DRIVE leaves the mutual-cooperation and mutual-defection payoffs unchanged and swaps the temptation and sucker payoffs under unilateral defection. Applying the same calculation to $(R', P', T', S')$ shows that the transformed payoffs obey the same pattern. In both cases, the resulting game has $(C, C)$ as its unique Nash equilibrium, so the dominance of cooperation is preserved. □

## A.10 Training with Normalized Returns

In our paper, $\hat{\pi}_i$ and $\hat{V}_i$ are trained with *normalized returns* $\overline{G}_{t,i} = \frac{1}{\sigma_i}(G_{t,i} - \mu_i)$, where $\mu_i$ and $\sigma_i$ are the mean and standard deviation of all returns $G_{t,i}$ of agent $i$ in an epoch $m$, to improve training stability (van Hasselt et al., 2016). The normalization makes standard policy gradient RL (without PI mechanism) invariant to reward changes, where rewards are scaled by a factor $c_m > 0$ or shifted by a scalar $b_m \in \mathbb{R}$:

**Lemma 7.** *Standard policy gradient RL according to Eq. 2, is invariant to reward changes, where the original environmental reward is scaled by a factor $c_m > 0$ or shifted by a scalar $b_m \in \mathbb{R}$, when the obtained discounted returns $G_{t,i} = \sum_{k=0}^{\infty} \gamma^k u_{t+k,i}$ in an epoch $m$ are normalized to zero mean and standard deviation such that $\overline{G}_{t,i} = \frac{1}{\sigma_i}(G_{t,i} - \mu_i)$.*

*Proof.* Let the transformed reward be $u'_{t,i} = c_m u_{t,i} + b_m$, with $c_m > 0$. Then the corresponding continuing discounted return is

$$G'_{t,i} = \sum_{k=0}^{\infty} \gamma^k(c_m u_{t+k,i} + b_m) = c_m \sum_{k=0}^{\infty} \gamma^k u_{t+k,i} + b_m \sum_{k=0}^{\infty} \gamma^k = c_m G_{t,i} + \frac{b_m}{1 - \gamma}. \tag{27}$$

Hence, every return in the epoch is transformed by the same positive affine map $G'_{t,i} = c_m G_{t,i} + B_m$, with $B_m := b_m/(1 - \gamma)$, independent of $t$. Therefore, the corresponding mean and standard deviation satisfy $\mu'_i = c_m \mu_i + B_m$ and $\sigma'_i = c_m \sigma_i$, and the normalized return becomes

$$\overline{G}'_{t,i} = \frac{G'_{t,i} - \mu'_i}{\sigma'_i} = \frac{c_m G_{t,i} + B_m - (c_m \mu_i + B_m)}{c_m \sigma_i} = \frac{G_{t,i} - \mu_i}{\sigma_i} = \overline{G}_{t,i}. \tag{28}$$

Thus, positive scaling and additive shifting of the reward do not affect the normalized return $\overline{G}_{t,i}$ and substituting $\overline{G}'_{t,i}$ into the policy gradient estimator Eq. (2) yields the same policy gradient.

$\square$

According to Lemma 7, the non-PI baselines *Naive Learning*, *LOLA-PG*, and *POLA-DiCE* are not affected by the dynamic reward changes in Sec. 8 and Fig. 12 (but are still less cooperative than *DRIVE*). However, most state-of-the-art PI methods like *LIO* and *MATE* cannot accommodate such changes (by the generally unknown function $f_{mod}$) in their payoff modifications (Figs. 2e, 2c) and thus fail to adapt, in contrast to *DRIVE*, as shown in Figs. 1c and 6.

### A.11 Invariance in SSDs Under Normalized Policy Gradients

We can restate and prove Theorem 4 in more detail using the normalized return defined in Lemma 7

$$\overline{G}_{t,i} = \frac{G_{t,i} - \mu_i}{\sigma_i},$$

where $\mu_i$ and $\sigma_i$ are the mean and standard deviation of all returns $G_{t,i}$ for agent $i$ in epoch $m$.

*Proof of Theorem 4.* Let the environmental rewards in epoch $m$ be transformed by $f_{mod}(r_{t,i}, m) = c_m r_{t,i} + b_m$ with $c_m > 0$. For any agent $i$, the DRIVE shaping rule in Eq. (7) depends only on $\hat{u}_{t,i}$ and differences of the form $\Delta_{t,i,j} = \bar{u}_j - \hat{u}_{t,i}$. Under $f_{mod}$ these become $\Delta'_{t,i,j} = \bar{u}'_j - \hat{u}'_{t,i} = (c_m \bar{u}_j + b_m) - (c_m \hat{u}_{t,i} + b_m) = c_m(\bar{u}_j - \hat{u}_{t,i}) = c_m \Delta_{t,i,j}$. The minima in Eq. (7) therefore scale by the same factor $c_m$, and the shaped reward transforms as

$$u_{t,i}^{\text{DRIVE}\,'} = c_m u_{t,i}^{\text{DRIVE}} + b_m,$$

i.e., $u_{t,i}^{\text{DRIVE}}$ is itself subjected to the same per-epoch positive affine map. Applying Lemma 7 to $u_{t,i}^{\text{DRIVE}}$ instead of $u_{t,i}$ shows that the normalized returns (and thus the policy updates) are unchanged. Hence, both the baseline policy-gradient dynamics and the incentive effects induced by DRIVE are invariant to per-epoch positive affine transformations, as claimed. $\square$

### A.12 Robustness to Stochastic Reward Perturbations

Definition 1 considered shared affine reward transformations and denoted the resulting rewards by $\hat{r}$. We now analyze a different source of variation: stochastic reward perturbations according to Definition 2: Let $\tilde{r}_{t,i} = r_{t,i} + \varepsilon_{m,i}$ denote the perturbed reward of the the environmental reward $r_{t,i}$ of agent $i$ at time $t$, with independent zero-mean Gaussian noise $\varepsilon_{m,i} \sim \mathcal{N}(0, \sigma^2)$ in epoch $m$. Unless stated otherwise, all perturbations are mutually independent and share the same variance $\sigma^2$.

For the following analysis, assume equally spaced PD payoffs $T > R > P > S$, $T - R = R - P = P - S = d$. The adjacent spacing $d$ controls how much stochastic perturbation can be tolerated before local payoff orderings are likely to change. The standard PD contains two individual incentives against cooperation: *greed*, $T > R$, and *fear*, $P > S$. As shown in Theorem 1 and Appendix A.1, DRIVE resolves both by swapping the unilateral incentives $T$ and $S$. After this swap, cooperation is favored against a cooperator if $R > S$, and cooperation is favored against a defector if $T > P$. These inequalities are automatically true under the original PD ordering. Repeated PDs additionally commonly assume the *welfare* condition $2R > T + S$, ensuring that mutual cooperation is socially preferable to alternating exploitation. Under equal spacing, $T = R + d$, and $S = R - 2d$, so $2R - (T + S) = d > 0$. Thus equal spacing preserves welfare optimality.

**Lemma 8** (Ordering probability). *For two independently perturbed rewards with gap $D = r - r'$, $\tilde{r} = r + \varepsilon$,*

$$\Pr[\tilde{r} > \tilde{r}'] = \Phi\left(\frac{D}{\sqrt{2}\sigma}\right),$$

*where $\Phi$ is the standard normal CDF.*

*Proof.* Since $\tilde{r}' = r' + \varepsilon'$ and $r - r' = D$, We have $\tilde{r} - \tilde{r}' = D + (\varepsilon - \varepsilon')$. Since $\varepsilon - \varepsilon' \sim \mathcal{N}(0, 2\sigma^2)$, it follows that $\Pr[\tilde{r} > \tilde{r}'] = \Pr[D + \eta > 0] = \Pr[\eta > -D]$, with $\eta \sim \mathcal{N}(0, 2\sigma^2)$. Standardizing yields $\Pr[\tilde{r} > \tilde{r}'] = \Phi\left(\frac{D}{\sqrt{2}\sigma}\right)$. $\square$

**Per-agent perturbations**   First consider the case where each agent draws one perturbation that is added uniformly to all payoff entries according to Definition 2: $\tilde{T}_i = T + \varepsilon_i$, $\tilde{R}_i = R + \varepsilon_i$, $\tilde{P}_i = P + \varepsilon_i$, $\tilde{S}_i = S + \varepsilon_i$. Then each agent's local ordering is preserved exactly: $T > R > P > S \implies \tilde{T}_i > \tilde{R}_i > \tilde{P}_i > \tilde{S}_i$. The only remaining failure mode is cross-agent inversion in exploitation states, for example $\tilde{T}_i \leq \tilde{S}_j$, because then the DRIVE response $\Delta_{t,i,j} = \bar{u}_j - \tilde{u}_{t,i}$ need not remain negative. The relevant distance is therefore $D = T - S = 3d$. Using Lemma 8, preserving this ordering with probability $\Phi(3) \approx 0.999$ yields

$$\frac{D}{\sqrt{2}\sigma} = 3 \quad \implies \quad \sigma_w = \frac{D}{3\sqrt{2}} = \frac{d}{\sqrt{2}}. \quad \text{Hence } \Pr[\tilde{T}_i > \tilde{S}_j] = \Phi(3) \approx 0.99865.$$

Thus, comparatively large per-agent perturbations can be tolerated because only cross-agent exploitation inversions matter.

**Independent payoff perturbation**   A more difficult case arises when each payoff is altered independently,

$$\tilde{T} = T + \varepsilon_T, \qquad \tilde{R} = R + \varepsilon_R, \qquad \tilde{P} = P + \varepsilon_P, \qquad \tilde{S} = S + \varepsilon_S,$$

with independent $\varepsilon_. \sim \mathcal{N}(0, \sigma^2)$. Then even the local PD ordering may change. The critical distances become adjacent payoff gaps of size $D = d$. To preserve adjacent orderings such as $\tilde{R} > \tilde{P}$ with probability $\Phi(3)$, Lemma 8 gives

$$\frac{d}{\sqrt{2}\sigma} = 3 \quad \implies \quad \sigma_n = \frac{d}{3\sqrt{2}}.$$

Figure 4 visualizes this transition. For $\sigma = d$, substantial overlap occurs and ordering violations become common. For $\sigma_w = d/\sqrt{2}$, the outer exploitation ordering $T > S$ remains highly robust while adjacent orderings still overlap noticeably. For the stricter scale $\sigma_n = d/3\sqrt{2}$, adjacent payoff orderings are preserved with probability approximately 0.999. Table 2 enumerates the dominant payoff permutations induced by independent perturbations under $\sigma_w$. The columns Greed, Fear, and Welfare denote the inequalities $T > R$, $P > S$, $2R > T + S$, respectively. The Gate column records whether $T$ remains the maximal payoff, corresponding to the regime where the TD-gating theorem directly applies. The upper block contains social dilemmas and cooperative games that remain solvable by DRIVE. The lower block contains brittle cases where either the TD-gating assumption fails or the resulting ordering becomes structurally anti-cooperative. Equality cases occur with probability zero under continuous Gaussian perturbations and are therefore omitted. In practice, ties can be broken by adding arbitrarily small deterministic offsets.

| Ordering | Mass $\sigma_w$ | Greed | Fear | Welfare | Gate | Interpretation |
|---|---|---|---|---|---|---|
| **Social dilemmas resolved by DRIVE and cooperative games** (0.836) | | | | | | |
| $T > R > P > S$ | 0.557 | ✓ | ✓ | ✓ | ✓ | DRIVE resolves greed and fear to favor mutual cooperation |
| $T > P > R > S$ | 0.133 | ✓ | ✓ | – | ✓ | DRIVE resolves greed and fear to favor mutual cooperation |
| $T > R > S > P$ | 0.121 | ✓ | – | ✓ | ✓ | Chicken; DRIVE resolves greed to favor mutual cooperation |
| $R > T > S > P$ | 0.024 | – | – | ✓ | – | Naive learning favors cooperation; DRIVE does not break it |
| $R > S > T > P$ | 0.001 | – | – | ✓ | – | Naive learning favors cooperation; DRIVE does not break it |
| **Brittle social dilemmas outside TD-gate guarantee and anti-cooperative cases** (0.164) | | | | | | |
| $R > T > P > S$ | 0.120 | – | ✓ | ✓ | – | TD gate does not activate; DRIVE cannot resolve fear |
| $R > P > T > S$ | 0.009 | – | ✓ | ✓ | – | TD gate does not activate; DRIVE cannot resolve fear |
| $P > T > R > S$ | 0.009 | ✓ | ✓ | – | – | TD gate does not activate; DRIVE cannot resolve fear |
| $P > R > T > S$ | 0.004 | – | ✓ | – | – | TD gate does not activate; DRIVE cannot resolve fear |
| $T > S > R > P$ | 0.009 | ✓ | – | ✓ | ✓ | TD gate activates but DRIVE swap cannot resolve greed |
| $T > P > S > R$ | 0.009 | ✓ | ✓ | – | ✓ | TD gate activates but DRIVE swap cannot resolve greed |
| $T > S > P > R$ | 0.004 | ✓ | – | – | ✓ | TD gate activates but DRIVE swap cannot resolve greed |

Table 2: Strict payoff orderings induced by independent Gaussian payoff-entry perturbations under equal spacing $d$. Each row analyzes one ordering permutation where the respective probability mass for $\sigma_w = d/\sqrt{2}$ is non-zero regarding the ordering's properties greed ($T > R$), fear ($P > S$), and welfare ($2R > T + S$), as well as *gate* ($T > \{R, S, P\}$), i.e., whether the TD gate opens for unilateral defection (cf. Theorem 1). Cases are split in two categories: scenarios that can be resolved by DRIVE to favor mutual cooperation (upper) and scenarios where DRIVE cannot resolve the remaining greed or fear condition (lower). Overall, even under $\sigma_w$, DRIVE resolves over 80% of the resulting situation towards mutual cooperation.

## B   Technical Details and Hyperparameters

### B.1   Hyperparameters

All common hyperparameters used by all MARL approaches in the experiments, as reported in Sec. 6 in the paper, are listed in Table 3. The final values were chosen based on a coarse grid search which finds a tradeoff between performance and computation w.r.t. *DRIVE, LIO, IA*, and *MATE* in *IPD* and *Coin-2* without dynamically changing rewards, i.e., $f_{mod}(u, m) = u$. We directly adopted the final values in Table 3 for all other domains. All training and test runs were performed in parallel on a computing cluster of 15 x86_64 GNU/Linux (Ubuntu 18.04.5 LTS) machines with i7-8700 @ 3.2GHz CPU (8 cores) and 64 GB RAM. We did not use any GPU in our experiments.

Table 3: Common hyperparameters and their respective final values used by all algorithms evaluated in the paper. We also list the numbers that have been tried during the development of the paper.

| Parameter | Final Value | Values/Range | Description |
|---|---|---|---|
| $K$ | 10 | $\{1, 5, 10, 20\}$ | Number of episodes per epoch. |
| $E$ | 4000 | $\{2000, 4000, 8000\}$ | Number of epochs. $E$ was gradually increased to assess the stability of the learning progress until convergence. |
| $\alpha$ | 0.001 | $\{0.001\}$ | Learning rate. We used the default ADAM implementation in `torch` without further tuning. |
| Clip norm | 1 | $\{1, \infty\}$ | Gradient clipping parameter. Using a clip norm of 1 leads to better performance than deactivating it with $\infty$. |
| $\lambda$ | 1 | $\{0, 1\}$ | Trace parameter for TD($\lambda$) critic learning. |
| $\gamma$ | 0.95 (*IPD, Coin*) 0.99 (*Harvest*) | $\{0.9, 0.95, 0.99\}$ | Discount factor for the return $G_{t,i}$. Any value $\geq 0.95$ would be sufficient. |
| $\|\tau_{t,i}\|$ | 1 | $\{1, 5, 10\}$ | Local history length. It was set to 1 to reduce computation because the other values did not significantly improve performance. |

For *LIO*, we set the cost weight for incentive function learning to 0.001 and the maximum incentive value $R_{\max}$ to the highest absolute penalty per domain (3 in *IPD*, 2 in *Coin-2* and *Coin-4*, and 0.25 in *Harvest-12*), as suggested in (Yang et al., 2020). For *IA*, we set $\alpha = 5$ and $\beta = 0.05$, as suggested in (Hughes et al., 2018).

### B.2   Neural Network Architectures

We coarsely tuned the neural network architectures in the paper w.r.t. performance and computation by varying the number of units per hidden layer $\{32, 64, 128\}$ for $\hat{\pi}_i$ and $\hat{V}_i$. The number of hidden layers was varied between 1, 2, and 3, but significantly deeper architectures led to deteriorating performance (possibly due to vanishing gradients). Using ELU or ReLU activation did not make any significant difference for any neural network. Thus, we stick to ELU throughout the experiments. *DRIVE, MATE*, and *Naive Learning* only use $\hat{\pi}_i$ and $\hat{V}_i$ as neural networks. The incentive function network of *LIO* has the same hidden layer architecture as $\hat{\pi}_i$ and $\hat{V}_i$. In addition, the joint action of the $n-1$ other agents is concatenated to the flattened observations before being input into the incentive function, which outputs an $n-1$ dimensional vector. The output vector is passed through a sigmoid function and multiplied with $R_{\max}$ (Sec. B.1) afterward.

### B.3   Coin-2 and Coin-4

We adopted the setup of (Foerster et al., 2018) in *Coin-2*, as shown in Fig. 7a, with the same rules and reward functions. The order of executed actions is randomized such that situations where two agents simultaneously step on a blue coin lead to an expected payoff of +1 for the red agent and -1 for the blue agent (Fig. 7a left), and vice versa for a red coin. In addition, we extended the domain to 4 agents in *Coin-4* (Fig. 7a right). All agents are able to move freely, and grid cell positions can be occupied by multiple agents. Any attempt to move out of bounds is treated as *do nothing* action.

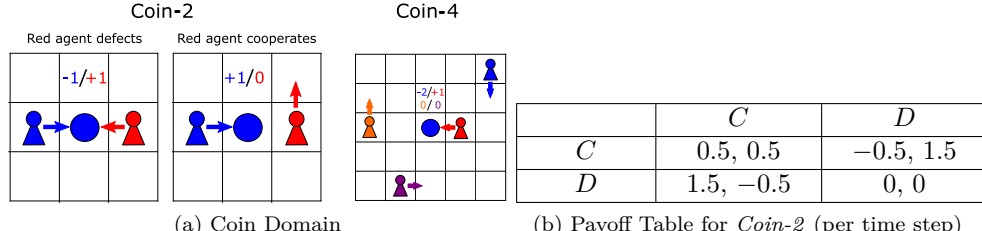

Figure 7: (a) *Coin-2* and *Coin-4*. (b) Payoff table for *Coin-2* w.r.t. the expected rewards per time step.

**Prisoner's Dilemma Connection**   Assuming that an agent always picks up a coin at every time step in *Coin-2*, the expected rewards per time step represent a PD instance, according to Sec. 2 and Fig. 1:

- If both agents cooperate, i.e., only collect their matching coins, each agent gets $R = 0.5(1 + 0) = 0.5$ on average.

- If both agents defect, i.e., collect any coin regardless of their color, each agent gets $P = 0.25(1+1-2+0) = 0$ on average.

- If one agent defects, the defecting agent gets $T = 0.5(1 + 0) + 1 = 1.5$ on average (its own coin 100% of the time and the other's coin 50% of the time). The exploited agent gets $S = 0.5(1 - 2) = -0.5$

The expected payoffs $R$, $P$, $T$, and $S$ satisfy the characteristic PD inequalities w.r.t. greed and fear, namely $T > R > P > S$. Thus, DRIVE also works well in *Coin-2* and *Coin-4* (which extends the SD to 4 agents), even with changing rewards, as shown in Fig. 6.

## B.4   Harvest-12

We adopted the setup of (Perolat et al., 2017a) in *Harvest-12*, as shown in Fig. 8a, with the same dynamics and apple regrowth rates. *Harvest-12* always starts with the initial apple configuration in Fig. 8a with randomly positioned agents. The agents can observe the environment within their $7 \times 7$ area and have no specific orientation. Thus, they have 4 separate actions to tag all neighbor agents, namely north, south, west, and east. All agents are able to move freely, and grid cell positions can be occupied by multiple agents. Any attempt to move out of bounds is treated as *do nothing* action.

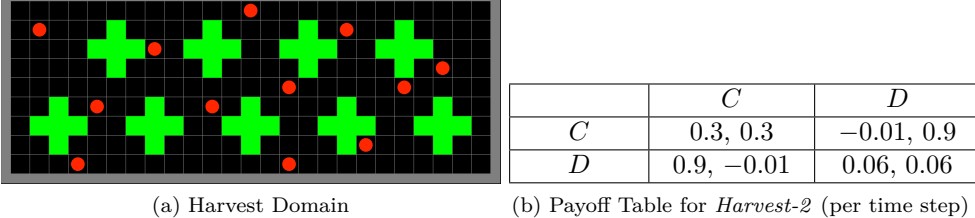

(a) Harvest Domain

|   | $C$ | $D$ |
|---|---|---|
| $C$ | 0.3, 0.3 | $-0.01$, 0.9 |
| $D$ | 0.9, $-0.01$ | 0.06, 0.06 |

(b) Payoff Table for *Harvest-2* (per time step)

Figure 8: (a) Domain layout with the initial apple configuration used for *Harvest-12*. (b) Payoff table for *Harvest-2* w.r.t. the expected rewards per time step.

**Prisoner's Dilemma Connection**   We empirically tested a 2-agent instance of Harvest to determine the payoff table, similar to (Leibo et al., 2017). The table is shown in Fig. 8b and represents a PD instance, according to Sec. 2 and Fig. 1:

- If both agents cooperate, i.e., no stunning and occasional waiting for apple regrowth, each agent gets $R \approx 0.3$ on average.

- If both agents defect, i.e., stunning and complete harvesting that prevents apple regrowth, each agent gets $P \approx 0.06$ on average.

- If one agent defects, i.e., stunning and harvesting, the defecting agent gets $T \approx 0.9$, and the exploited agent gets $S \approx -0.01$ on average (due to being stunned most of the time with a 0.01 time penalty).

The expected payoffs $R$, $P$, $T$, and $S$ satisfy the characteristic PD inequalities w.r.t. greed and fear, namely $T > R > P > S$. Thus, DRIVE also works well in *Harvest-12* (which extends the SD to 12 agents), even with changing rewards, as shown in Fig. 6 and Fig. 13. This also confirms the empirical result of (Leibo et al., 2017), stating that Harvest can represent a PD instance w.r.t. greed and fear, depending on the availability and regrowth rate of apples.

## C   Additional Results

Unless stated otherwise, the following additional experiments are preliminary and were conducted under the constrained rebuttal timeline. They are averaged over 10 runs and trained for 1000 epochs, rather than the 20 runs and 4000 epochs used in the main experiments. We focus on the unchanged reward setting and the two representative reward-change schedules $f_{mod}^{I}$ and $f_{mod}^{IV}$, corresponding to linear increase and damped cosine modulation, respectively.

### C.1   Stochastic Reward perturbations

To complement the theoretical analysis of stochastic perturbations in Appendix A.12, we evaluate DRIVE under per-agent reward noise in the IPD and *Coin-2*. We scale the environmental rewards by a factor of 5, matching the motivating example in Fig. 1c, and add independent per-agent, per-epoch Gaussian noise with $\sigma = 5/\sqrt{2}$. This corresponds to the wider perturbation scale analyzed theoretically, where the exploitation ordering $T > S$ is preserved with high probability while individual agents may experience different reward offsets.

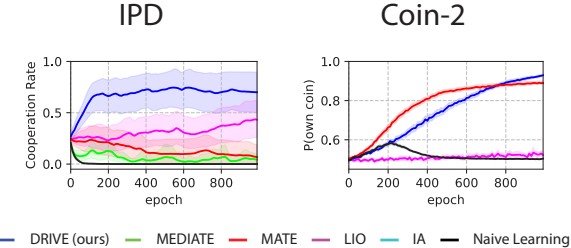

Figure 9: Preliminary empirical results under per-agent stochastic reward perturbations in IPD and *Coin-2*. Environmental rewards are scaled by a factor of 5 and perturbed by independent per-agent, per-epoch Gaussian noise with $\sigma = 5/\sqrt{2}$. Results are averaged over 10 runs and trained for 1000 epochs. DRIVE maintains cooperative behavior under these perturbations and achieves the strongest overall cooperation among the compared methods, indicating that the reward-difference exchange remains effective when the underlying social-dilemma ordering is preserved with high probability.

Fig. 9 shows that DRIVE remains robust under this form of agent-specific reward noise. In the IPD, DRIVE quickly reaches and maintains high cooperation despite the perturbed reward scale. In *Coin-2*, DRIVE likewise preserves a high own-coin rate, whereas the baselines are more strongly affected by the noisy reward perturbations. These results support the theoretical intuition that DRIVE does not require exact affine reward transformations, but rather depends on the preservation of the relevant social-dilemma ordering.

### C.2   Mean-reward ablations

DRIVE uses the responder's epoch-average reward when computing the response term $\Delta$. This design is intended to make the response sensitive to systematic disadvantage rather than to single noisy outcomes. To analyze the effect of this averaging window, we compare the original DRIVE variant using 10 episodes per epoch to two ablations: *DRIVE-short*, which uses only 2 episodes per epoch, and *DRIVE-instant*, which replaces the epoch-average response with an instantaneous reward response.

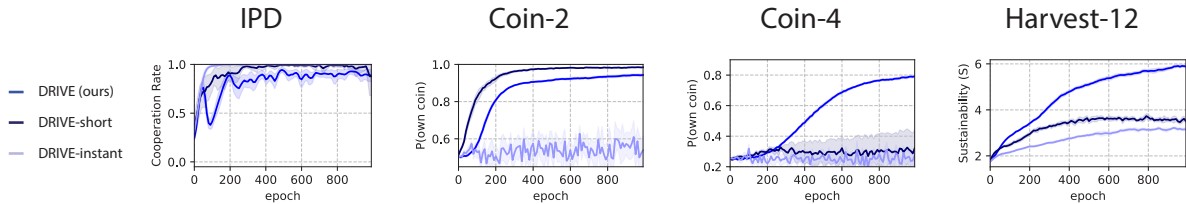

Figure 10: Ablation of the average-reward response mechanism. We compare the original DRIVE setup using a 10-episode mean reward response to *DRIVE-short*, which uses a 2-episode mean reward response, and *DRIVE-instant*, which responds using the instantaneous reward. Results are averaged over 10 runs and trained for 1000 epochs. Shorter or instantaneous estimates can be sufficient in simpler two-agent settings, but the longer averaging window becomes more robust as the number of agents and environmental complexity increase.

Fig. 10 shows that the effect of the averaging window depends on the complexity of the environment. In the two-agent IPD, instantaneous responses can perform well because the reward signal is dense and directly aligned with the current interaction. In *Coin-2*, shorter epochs mainly slow convergence. In larger or more temporally extended environments such as *Coin-4* and *Harvest-12*, the 10-episode average is more stable, suggesting that averaging helps filter temporal variance and improves the alignment between exploitative behavior and the resulting response.

### C.3    Additional baselines

To address the coverage of recent peer-incentivization baselines, we additionally evaluate MEDIATE under representative reward-change settings. MEDIATE adapts token magnitudes through local endorsement and consensus-like updates, making it particularly relevant for comparison with DRIVE under changing reward scales.

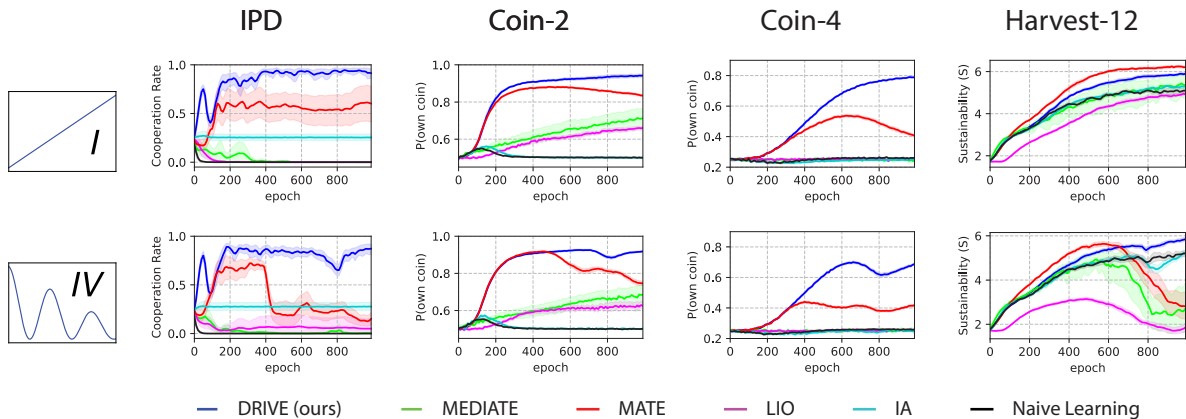

Figure 11: Preliminary MEDIATE baseline results under the linear increase $f^I_{mod}$ and damped cosine modulation $f^{IV}_{mod}$ reward-change schedules. MEDIATE was trained for 1000 epochs and averaged over 10 seeds. The initial results indicate that MEDIATE does not adapt robustly to the changing reward scales in these settings, presumably due to delayed update dynamics of its adaptive token mechanism. Full MEDIATE and LToS evaluations with the same experimental budget as the remaining baselines are planned for the final revision.

Fig. 11 shows that MEDIATE is less robust to the considered reward changes than DRIVE. This is most visible under $f^{IV}_{mod}$, where reward magnitudes change non-monotonically and occasionally approach low-scale regimes. The results suggest that mechanisms relying on learned or updated incentive magnitudes can lag behind reward-scale changes, whereas DRIVE adapts immediately because its incentive magnitude is computed directly from observed reward differences.

### C.4 Additional cooperation measures in *Harvest-12*

We provide additional results regarding different cooperation measures for *Harvest-12*, namely *Social Welfare or Efficiency (U)* (according to Sec. 2), *Equality (E)* (1 minus the Gini coefficient), *Sustainability (S)* (the average number of time steps at which apples are collected), and *Peace (P)* (the average number of untagged agents at any time step). All measures are based on the *original environmental rewards* $u_{t,i}$ instead of $\hat{u}_{t,i} = f_{mod}(u_{t,i}, m)$ (Algorithm 1, Line 15) to assess the stability of cooperation reliably. They are defined by (Perolat et al., 2017b):

$$U = \sum_{i \in \mathcal{D}} \sum_{t=0}^{H-1} u_{t,i}, \qquad\qquad E = 1 - \frac{\sum_{i \in \mathcal{D}} \sum_{j \in \mathcal{D}} |\sum_{t=0}^{H-1}(u_{t,i} - u_{t,j})|}{2n \sum_{i \in \mathcal{D}} \sum_{t=0}^{H-1} u_{t,i}},$$

$$S = \frac{1}{n} \sum_{i \in \mathcal{D}} \chi_i, \text{ where } \chi_i = \mathbb{E}[t|u_{t,i} > 0], \qquad P = n - \frac{1}{H} \sum_{i \in \mathcal{D}} \sum_{t=1}^{H} \mathbf{1}[\text{agent timed-out on time step } t]$$

Fig. 13 shows the progress of the alternative cooperation measures in *Harvest-12* with the different reward change functions $f_{mod}$ I, II, III, and IV (Fig. 12) from Sec. 8. While DRIVE is robust across all measures, MATE and LIO change significantly in most settings, except for equality in the reward change functions $f_{mod}^{I}$ and $f_{mod}^{III}$. IA is only affected by $f_{mod}^{IV}$ regarding all cooperation measures. Apart from $f_{mod}^{III}$, MATE always exhibits more peace than DRIVE, although the peace level significantly varies depending on the reward change function. LIO only exhibits more peace than DRIVE under $f_{mod}^{II}$ and $f_{mod}^{IV}$, which both converge to zero over time.

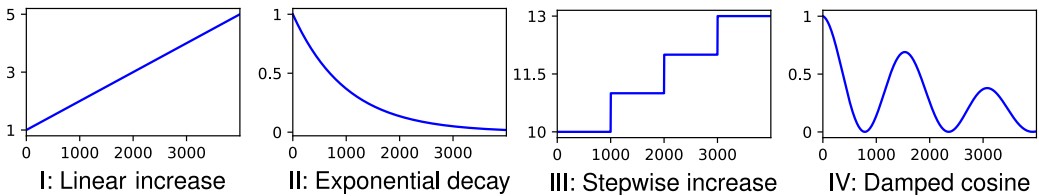

Figure 12: The reward change functions $f_{mod}$ used in each epoch $m$ according to Line 15 in Algorithm 1.

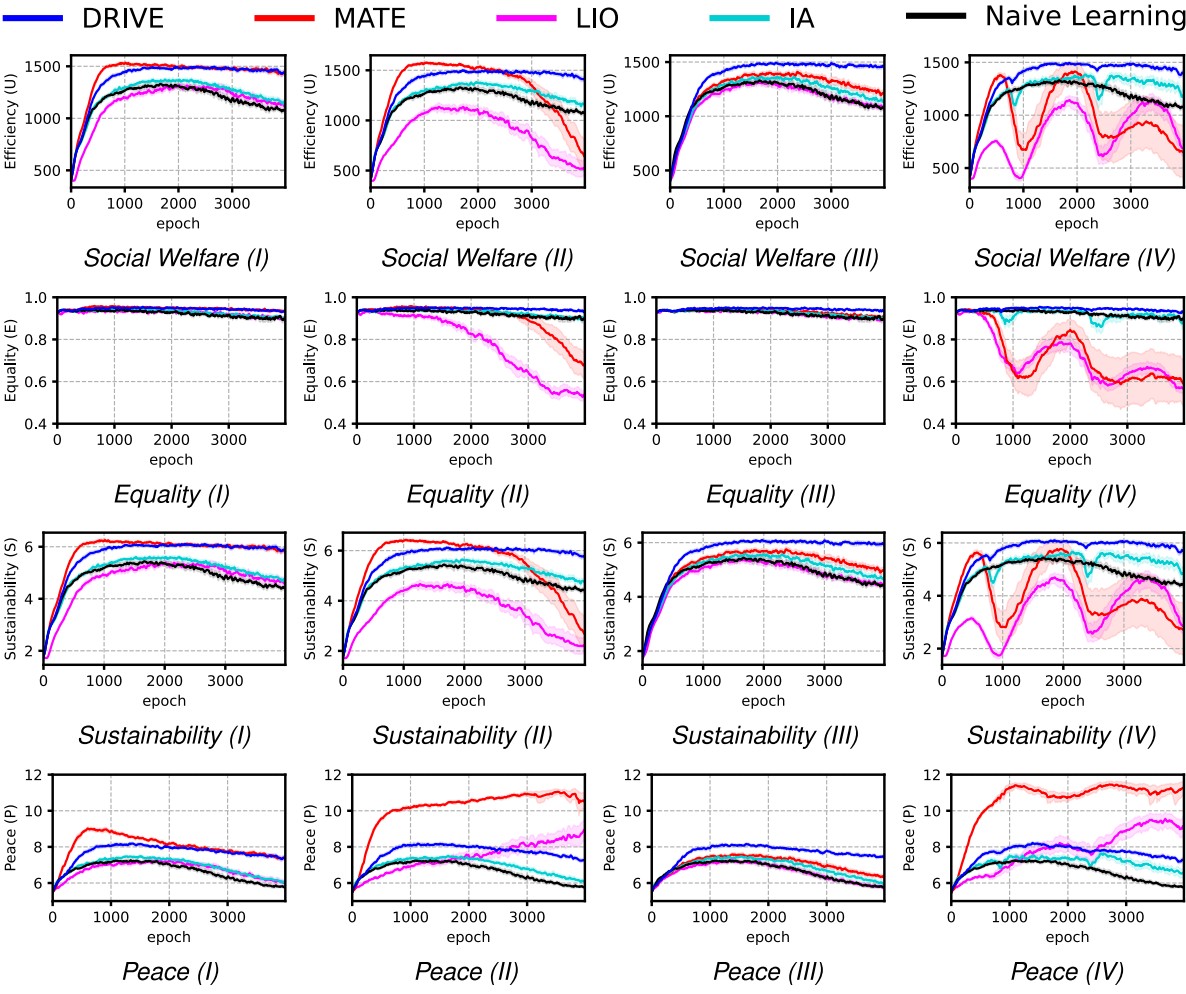

Figure 13: Average progress of DRIVE and other baselines in *Harvest-12* with different reward change functions $f_{mod}$ (I, II, III, and IV), as illustrated in the legend at the top and Fig. 12 regarding the cooperation measures *Social Welfare or Efficiency (U)*, *Equality (E)*, *Sustainability (S)*, and *Peace (P)* (Perolat et al., 2017b). Shaded areas show the 95% confidence interval.

