# OpenReview forum: "Dynamic Reward Incentives for Emergent Cooperation under Changing Rewards"
_TMLR — Decision pending for TMLR_

### Review · Reviewer_MjRa · 2026-04-20

**Summary Of Contributions:**

This work addresses the problem of learning optimal coordination behavior in (sequential) social dilemmas when the learning is decentralized and independent over the multiple agents comprising the system. To do so, the authors develop DRIVE, a new peer incentivization mechanism that incorporates with common reinforcement learning techniques such as policy gradient.

The paper demonstrates the capacity of DRIVE to reshape agents’ individual incentives in order to remove the un-optimal equilibria of general prisoner’s dilemmas. Experimentally, DRIVE shows strong performance on a range of more complex (sequential) social dilemmas. Compared to previous algorithms, DRIVE maintains its performance under per-epoch reward transformations.

Strengths:

- The problem is well introduce and the overall method well explained
- The experiments appear sound and complement the theoretical motivations of the algorithm
- DRIVE is shown to handle time-dependent reward transformations, which is of great importance in practice

Weaknesses:

- The theoretical analysis of the work is limited in terms of convergence guarantees (none provided), reward transformations (affine transformations only) and the impact of environment parameters (neighborhood size, variance)

**Audience:**

Yes

**Audience Explanation:**

Incentivizing collaboration in decentralized learning approaches is highly relevant to TMLR’s audience. The paper presents a simple and elegant peer-incentivization mechanism that is proven robust to affine transformations of the rewards.

**Broader Impact Concerns:**

Nothing to be mentioned in particular.

**Claims And Evidence:**

Yes

**Claims Explanation:**

The theoretical results, although relatively limited, seem sound. Regarding the experiments, the developed method appears to be successful in a variety of environment with varying reward transformations. See my comment below and making sure the comparisons are fair with proper hyperparameter tuning.

**Requested Changes:**

Overall, I find the paper well written and the theory and experiments appear sound. I believe that some additional clarifications would make the work even stronger.

**Questions and remarks:**

- **On convergence guarantees.** Page 7: “we emphasize that the mechanism does not guarantee convergence of learning dynamics, but rather reshapes incentives such that cooperation becomes the individually rational choice in repeated interactions”
This is a weakness of the work that could benefit from a mention in the final conclusion.
    - It would be interesting to understand the challenges in proving the convergence of the proposed method. Does the convergence behavior reduce to the properties of the specific learning algorithms in use?
    - In the repeated static game setting, would it be possible to apply bandit algorithms and show online regret guarantees? Or would a metric different from regret be more interesting for the problem considered in the paper?
- **On the motivation of the decentralized approach.** The paper could benefit from a short explanation on why the problem cannot be naively solved by training on a single reward corresponding to the social welfare in an identical interest game. The motivation seems to be similar to the motivation for defining neighbors, modeling a distributed systems where inter-agent communication is limited.
- **Impact of neighborhood size.** How does the definition of the neighbors influence the performance of the algorithm? Is having more neighbors always beneficial in terms of convergence speed?
- **Impact of variance in dynamic games.** On page 3: “SSDs enable more realistic scenarios in which behavior is captured by policies rather than by atomic actions. These can still be mapped to matrix games by classifying policies as C or D and evaluating empirical payoffs (Leibo et al., 2017), making core SD concepts applicable to SSDs”
    - I agree with the authors that such reductions make it possible to motivate DRIVE in the setting of SSDs.
    - In the case of dynamic games, we might experience high variance in the individual returns of the players, what is the impact of such variance? For example, on the convergence speed of the algorithm.
- **Baseline hyperparameters.** As mentioned by the authors, previous methods’ performance can be quite sensitive to their hyperparameters and require additional tuning when the rewards are transformed. Page 10 it is mentioned “We use LIO, MATE with x= 1 […], and IA with alpha = 5 and beta = 0.05 […]”, these values being suggesting by previous work. The paper tends to show that DRIVE outperforms all previous methods across diverse environments. The reviewer wonders how fair these comparisons are when hyperparameters haven’t been tuned to specific evaluation environments.

**Minor adjustments / formatting:**

- On page 3: “often leading to overly greedy behavior and mutual defection unless rewards or incentives are modified”.
Please give a reference for this claim.
- On the notation for partially observable Markov Games.
    - The notation seems a little unusual, and could be clarified.
    - From the presentation of S and A, they seem to represent trajectories because of the time indices, whereas I suppose they represent the state and action spaces, respectively.
    - Similarly, the definition of the value function can be confusing, because of the implicit dependence on the time step t through the state s_t instead of V_t. I believe there is also a missing \ before the “max” in the definition of the optimal value.
- Missing \ before log and missing _, for Equation (2). Also missing some \ before the min in Algorithm 2. These are minor LaTeX related suggestions, feel free to ignore.

---

> ### Author Response · Authors · 2026-05-27
> **Rebuttal (1/2)**
>
> We thank the reviewer for the constructive assessment and helpful suggestions. We appreciate the positive comments on the motivation, clarity, and empirical evaluation. We have revised the manuscript accordingly, with changes marked in blue.
>
> ## Convergence guarantees
>
> We agree that DRIVE does not provide a full convergence guarantee for general actor-critic learning dynamics in SSDs. The revised manuscript now states this more prominently in the theory and limitations sections. We clarify that the central theoretical results should be interpreted as reward-level incentive-alignment guarantees, not as convergence guarantees for arbitrary MARL learners in temporally extended environments.
>
> To strengthen the convergence intuition in the repeated static-game setting, we added a continuous mean-field analysis for the two-player PD. Policies are parameterized by cooperation probabilities. The unshaped PD induces local gradients toward defection for all opponent cooperation rates because of greed and fear, whereas DRIVE reshapes the reward surface so that the induced vector field points toward mutual cooperation. The revised manuscript includes a new figure visualizing the unshaped and DRIVE-shaped reward surfaces and vector fields.
>
> We also added a finite-grid no-regret corollary. By discretizing cooperation probabilities, the DRIVE-shaped game has full cooperation as the unique Nash equilibrium and the unique coarse correlated equilibrium under the mean-field assumptions. Standard finite-game no-regret results then imply empirical play concentrates on full cooperation. This does not prove convergence of the full actor-critic SSD implementation. The revised manuscript explicitly states that a full proof would additionally require critic calibration, concentration of running reward averages, and stochastic actor updates tracking the limiting DRIVE-shaped vector field despite simultaneous learning and temporal credit assignment.
>
> ## Motivation for the decentralized approach
>
> We agree that the manuscript should explain why one cannot simply optimize a shared social-welfare reward. We revised the introduction to clarify that our target setting is decentralized and ad hoc: agents may be controlled by different stakeholders, manufacturers, or organizations and therefore cannot realistically rely on a centrally imposed global reward signal. DRIVE is designed for this setting: agents remain self-interested, use independent learning updates, and exchange only local incentive information with neighbors. This distinguishes DRIVE from identical-interest settings with a shared global reward.
>
> ## Impact of neighborhood size
>
> We agree that neighborhood size is an important modeling parameter. We expanded the theoretical and conceptual discussion around neighborhoods. The revision now explains that larger neighborhoods are not automatically beneficial in all respects. They can improve redundancy because, under DRIVE’s min aggregation, a defector can be penalized as soon as at least one exploited compliant neighbor responds truthfully. However, once sufficient compliant coverage exists, additional neighbors mainly add redundant exchanges rather than qualitatively changing the incentive structure.
>
> The revised appendix connects this to a graphical N-agent PD extension: pairwise DRIVE incentive alignment propagates through sufficiently connected communication graphs, and defector penalization is guaranteed when compliant agents form a dominating set. We also make clear that a systematic neighborhood-size sweep remains future work.
>
> ## Variance in dynamic games and epoch-average estimation
>
> We agree that high variance in SSD returns can affect incentive signals and the speed or stability of learning. We revised the discussion to clarify that matrix-game reductions of Coin and Harvest motivate DRIVE in SSDs, but do not fully capture the full temporally extended learning problem. The actual environments involve longer horizons, simultaneous-learning non-stationarity, temporal credit assignment, critic approximation error, and delayed temporal alignment between actions, reward differences, and incentive exchange.
>
> To evaluate the role of epoch-average estimation, we added an ablation comparing the original 10-episode mean response with a 2-episode response and an instantaneous-response variant. The results show that shorter or instantaneous estimates can work in simpler two-agent settings, but the longer averaging window is more robust as the number of agents and environment complexity increase. In Coin-2, shorter epochs mainly slow convergence; in Coin-4 and Harvest-12, the longer average is more stable.

---

> ### Author Response · Authors · 2026-05-27
> **Rebuttal (2/2)**
>
> ## Baseline hyperparameters
>
> We agree that the fairness of baseline comparisons should be stated more clearly. We revised the experimental setup to clarify that baseline hyperparameters follow prior studies that tuned the methods on the same or comparable environments. This choice is intentional: our main question is whether fixed or previously tuned PI mechanisms remain robust when reward scales change without retuning. DRIVE does not require an incentive magnitude or reward-scale-specific retuning.
>
> We also now distinguish this no-retuning robustness question from the different question of whether each baseline could be rescued by oracle retuning for every known reward-change schedule. We do not claim that retuning could never improve baselines under known transformations; rather, we evaluate robustness in an online/ad hoc setting where reward changes may be unknown.
>
> ## Minor clarifications and formatting
>
> We addressed the requested notation and formatting clarifications. The POMG notation has been clarified, the value-function notation has been revised, Equation (2) now includes the missing log, and the statement about independent learners in SDs leading to greedy behavior or mutual defection now includes references.

---

### Review · Reviewer_S16o · 2026-04-26

**Summary Of Contributions:**

DRIVE introduces a reciprocal reward exchange mechanism where agents share reward differences rather than fixed tokens, making the incentive scheme naturally adaptive to whatever reward scale the environment currently presents. The authors formally prove that this mechanism aligns agent incentives toward mutual cooperation in the general Prisoner's Dilemma by effectively swapping the temptation and sucker payoffs under unilateral defection, removing both greed and fear as obstacles to cooperation. Crucially, because DRIVE operates on differences rather than absolute values, it is mathematically invariant to affine reward transformations, unlike competing methods such as MATE or LIO whose fixed or learned incentive magnitudes are tied to the absolute reward scale. This invariance is extended theoretically to sequential social dilemmas when combined with normalized policy gradient training. Empirically, DRIVE is validated across four environments and four qualitatively different reward change schedules, consistently outperforming state-of-the-art peer incentivization baselines while requiring no hyperparameter retuning as rewards drift.

Strengths:

1. Well-motivated problem with practical relevance.
2. Theory and experiments are tightly aligned.
3. DRIVE requires no learned incentive function, no fixed hyperparameters, and no extended action space.
4. The paper clearly acknowledges where DRIVE can break down.

Weaknesses (more details in the requested changes below):

1.  Truthful communication is a strong and largely unverified assumption.
2. The theoretical analysis is largely restricted to the two-agent Prisoner's Dilemma.
3. The affine reward transformation class may be too restrictive.
4. Limited baseline diversity.
5. Sensitivity to epoch-average reward estimation.

**Additional Comments:**

N/A

**Audience:**

Yes

**Audience Explanation:**

The paper's results will be of interest to the MARL community.

**Claims And Evidence:**

No

**Claims Explanation:**

There are some significant limitations that I have listed in the weaknesses above and elaborated below in "Requested Changes". Most importantly, the baseline coverage for the experiments is insufficient missing the most related algorithms.

**Requested Changes:**

Requested Changes:

I am expanding on my weakness list above along with requesting changes.

1. **Truthful communication is a strong and largely unverified assumption**. The entire mechanism hinges on agents honestly reporting their rewards and epoch-average returns. While the paper acknowledges this and analyzes compliance cases, it does not propose or evaluate any concrete mechanism to enforce or incentivize truthful reporting. In realistic multi-agent deployments, strategic misreporting could be a significant practical concern that remains unaddressed experimentally. An experiment or at-least a discussion in the paper for this point is required.

2. **The theoretical analysis is largely restricted to the two-agent Prisoner's Dilemma**. While Theorem 4 extends invariance to SSDs and Appendix A.6 sketches an N-agent graphical PD extension, the core incentive alignment guarantees (Theorem 1) rely heavily on the clean two-agent PD structure. The gap between this idealized setting and the complex multi-agent environments tested empirically (e.g., Harvest-12 with 12 agents) is wide and not fully bridged theoretically. This needs to be acknowledged more readily in the paper, with additional discussions under limitations.

3. **The affine reward transformation class may be too restrictive**. Definition 1 assumes rewards change via a shared, uniform, positive affine map applied identically to all agents within an epoch. This excludes many realistic reward perturbations such as agent-specific noise, non-linear distortions, clipping, saturation, or state-dependent transformations. The paper acknowledges this limitation briefly in Appendix A.7 but does not evaluate DRIVE under such conditions, leaving its robustness in messier real-world settings unclear. A more elaborate discussion and/or experiments on this aspect is required.

4. **Limited baseline diversity**. The empirical comparisons focus on a relatively narrow set of peer incentivization baselines (LIO, MATE, IA), plus LOLA-PG and POLA-DiCE which are included only for two domains and taken directly from prior reported results rather than re-run under the same conditions. Notably absent are more recent methods such as LToS or MEDIATE, the latter of which is cited as directly related work addressing similar robustness concerns, making it a conspicuous omission from the experimental comparison. The authors should compare these methods or provide concrete reasons for their omission.

5. **Sensitivity to epoch-average reward estimation**. DRIVE's response mechanism depends on a running epoch-average reward that requires sufficient time within an epoch to become a reliable signal. Early in each epoch, or in short episodes, this average may be poorly estimated, potentially generating noisy or incorrect incentive signals. The paper does not systematically analyze how epoch length, estimation quality, or the cold-start phase affect cooperation dynamics, which is an important practical consideration for deployment. Additional experimentation and associated discussions are required to address these aspects.

---

> ### Author Response · Authors · 2026-05-27
> **Rebuttal (1/2)**
>
> We thank the reviewer for the detailed and constructive feedback. We appreciate the positive assessment of DRIVE’s motivation, theoretical alignment, and practical simplicity. We have revised the manuscript accordingly, with additions marked in blue.
>
> ## Truthful communication
>
> We agree that truthful communication is a strong assumption. The original submission already discussed non-adherence cases, but we agree that the practical implications and the absence of an enforcement mechanism needed to be made more explicit. In the revised manuscript, we now state in the main limitations that DRIVE does not currently enforce truthful reporting of rewards or reward differences. We explicitly acknowledge that strategic misreporting can distort the min aggregation, remove intended penalties, or introduce unjustified penalties when compliant responders are sparse. We also expanded the future-work discussion to include redundant aggregation, stochastic auditing, consensus-style validation, explicit incentives for truthful reporting, and agent identification as possible safeguards.
>
> At the same time, we clarified the graceful-degradation argument. Because DRIVE aggregates responses using a minimum, a defector can still be penalized whenever at least one exploited compliant neighbor responds truthfully. The revised theoretical discussion now connects this to communication topology: in graphical N-agent PDs, defector penalization is guaranteed when compliant agents form a dominating set, so partial non-compliance weakens but does not necessarily collapse the incentive structure. We agree that this does not solve adversarial reporting, and we now explicitly identify dedicated adversarial-reporting experiments as future work.
>
> ## Scope of the theory
>
> We agree that the strongest formal guarantee is in the clean two-agent PD. We revised the manuscript to distinguish more clearly between reward-level incentive alignment and convergence guarantees for full actor-critic MARL in SSDs. The revised discussion now states that Coin and Harvest can be abstracted into matrix games that preserve the required dilemma structure, but that these abstractions do not capture all difficulties of the temporally extended environments, including longer horizons, simultaneous-learning non-stationarity, critic approximation error, temporal credit assignment, and delayed alignment between actions, reward differences, and incentive exchange.
>
> To strengthen the theoretical intuition, we added a continuous mean-field analysis for the two-player PD. In this analysis, policies are parameterized by cooperation probabilities. The unshaped PD induces gradients toward defection, whereas the DRIVE-shaped mean-field reward redirects the vector field toward mutual cooperation. We also added a finite-grid no-regret corollary. These results do not prove convergence of the full actor-critic implementation in SSDs, and the revised manuscript explicitly says so, but they make the repeated-game convergence intuition more precise.
>
> ## Reward transformations beyond affine changes
>
>  We agree that shared positive affine transformations cover only a tractable subset of possible reward perturbations. We therefore added a new theoretical and empirical section on stochastic reward perturbations. The revised manuscript now analyzes Gaussian perturbations and derives conditions under which the relevant social-dilemma ordering is preserved with high probability. For per-agent perturbations, the local payoff ordering remains intact and the critical failure mode is a cross-agent inversion between exploitation and sucker outcomes. For independent payoff perturbations, adjacent payoff orderings must also be preserved, which yields a stricter noise-scale condition. The revised manuscript also includes a visualization of noisy payoff overlap and preliminary experiments in IPD and Coin-2 under per-agent Gaussian reward noise. These results indicate that DRIVE remains the most cooperative evaluated method when the relevant social-dilemma ordering is preserved with high probability.
>
> We also clarify that this still does not cover all non-affine changes. In particular, clipping, saturation, sign-dependent remapping, and state-dependent transformations remain outside the current theoretical guarantee and are now explicitly framed as future work.

---

> ### Author Response · Authors · 2026-05-27
> **Rebuttal (2/2)**
>
> ## Baseline diversity
>
> We agree that the original baseline coverage was limited with respect to recent methods such as LToS and MEDIATE. In the revised manuscript, we expanded the related-work discussion to position LToS and MEDIATE more clearly. We also added preliminary MEDIATE experiments in Appendix C.3 for representative reward-change schedules, including linear increase and damped cosine modulation. These runs are explicitly marked as preliminary, using 10 seeds and 1000 epochs rather than the 20 seeds and 4000 epochs used in the main experiments. The preliminary results suggest that MEDIATE is less robust than DRIVE under these reward changes, especially when reward magnitudes vary non-monotonically.
>
> We acknowledge that the current revision strengthens the baseline discussion and adds preliminary MEDIATE evidence, but does not yet provide full-budget MEDIATE and LToS comparisons across all scenarios. We are committed to including full comparisons for both MEDIATE and LToS in all scenarios covered in the main paper in the final revision. This will allow us to assess these adaptive peer-incentivization methods under the same experimental budget and reward-change settings as the other baselines.
>
> ## Sensitivity to epoch-average reward estimation
>
> We agree that the epoch-average response can introduce cold-start and estimation-quality issues. We added an empirical ablation in Appendix C.2 comparing the original 10-episode mean response, a 2-episode mean response, and an instantaneous-response variant. The results show that instantaneous or short-window responses can work in simpler two-agent settings, while the longer averaging window becomes more stable in larger or more temporally extended environments such as Coin-4 and Harvest-12. This supports the design choice that the responder’s average reward filters temporal variance and makes DRIVE respond to systematic disadvantage rather than single noisy outcomes.
>
> Overall, the revised manuscript now makes the communication assumptions and theoretical scope more explicit, adds stochastic-perturbation analysis and preliminary noisy-reward experiments, adds preliminary MEDIATE results, and includes averaging-window ablations.

---

> > ### Comment · Reviewer_S16o · 2026-05-28
> > **Reply to the authors**
> >
> > I thank the authors for the detailed rebuttal and the revisions to their manuscript. Most of my concerns have been addressed. I encourage the authors to make sure to expand the preliminary experiments and provide complete results for the final manuscript in line with their responses above.

---

### Review · Reviewer_xv4n · 2026-05-13

**Summary Of Contributions:**

This paper proposes and evaluates a new peer incentivization (PI) method for promoting cooperation in multi-agent reinforcement learning under social dilemma settings.
The proposed method is based on the idea of exchanging rewards between agents under appropriate conditions, and is designed to operate adaptively even in situations where rewards change.
The effectiveness of the proposed method is verified through numerical experiments. In addition, desirable properties such as invariance under scale transformations are also established theoretically.

**Audience:**

Yes

**Audience Explanation:**

To be honest, since this is quite far from my area of expertise, I am not confident in estimating how experts in the field would receive this paper.
Nevertheless, as an outsider, I personally found the significance of the problem addressed in this paper convincing, and was able to read it with interest.
I also understand it as research that addresses an important problem that has been studied in many references, following recent developments in machine learning and AI.

**Claims And Evidence:**

Yes

**Claims Explanation:**

I am not an expert in this field and am not familiar with the line of prior work, so my assessment may not be appropriate by the standards of the community.

As far as I can tell from the numerical experiments, the data sufficiently support the claims made in the paper.
However, the proposed protocol appears to be designed specifically for the particular purpose of addressing social dilemmas, and I have some concerns that, depending on real-world application scenarios, it might exhibit unexpected side effects or potential fragility. That said, since the appendix includes robustness evaluation experiments addressing such concerns, and the main text also mentions the potential fragility, I do not consider this to fall under "lack of evidence."

Regarding the theoretical analysis, the results cannot be said to be highly non-trivial or surprising (in particular, Lemma 2 seems almost obvious, and I did not see why it needed to be stated as a lemma), but I was not able to find any errors.
However, regarding Theorem 2, while I am convinced of the statement of the theorem itself, I am not convinced that the claim is reasonable. That is, when scaling $f_{mod}$, isn't it more natural to scale the parameter $x$ simultaneously as well? If we restrict to such situations, my understanding is that the MATE protocol is also invariant.

**Requested Changes:**

I have no particular changes I would like made to the paper, but I would appreciate it if the authors could point out any misunderstandings on my part or respond to the concerns described above.

---

> ### Author Response · Authors · 2026-05-27
> **Rebuttal**
>
> We thank the reviewer for the careful reading and constructive comments. We appreciate the positive assessment of the motivation, numerical evidence, and theoretical correctness. We also appreciate the opportunity to clarify the intended scope of DRIVE and the interpretation of the MATE comparison.
>
> ## Scope of DRIVE and possible side effects
>
> We agree that DRIVE is specifically designed for social dilemmas, and we have clarified this scope more explicitly in the revised manuscript. DRIVE targets settings where individual and collective incentives are misaligned through greed and fear, such as Prisoner’s Dilemma-like social dilemmas. In such settings, unilateral defection is individually attractive even though mutual cooperation is socially preferable. The DRIVE payoff exchange is designed to remove exactly this incentive misalignment by reversing the temptation and sucker outcomes under unilateral defection.
>
> The revised manuscript now makes clearer that DRIVE should not be interpreted as a universal reward-shaping mechanism for arbitrary games. If a task is already fully cooperative, peer incentivization is not necessary; if reward perturbations destroy the relevant social-dilemma ordering, or if the task is not a social dilemma, the theoretical guarantees no longer apply. We also expanded the limitations and future-work discussion to state that extending the analysis beyond PD-like inequalities to broader classes of social dilemmas remains future work.
>
> In response to related reviewer comments, we also added a stochastic reward-perturbation analysis. This makes the fragility condition more explicit: DRIVE remains robust when the relevant social-dilemma ordering is preserved with high probability, but it should not be expected to preserve the same behavior once perturbations alter the underlying strategic structure.
>
> ## Interpretation of MATE under reward scaling
>
> We thank the reviewer for this insightful point. We agree that if the reward-scaling function were known and externally synchronized, one could scale MATE’s token parameter at the same time. Under such oracle retuning, MATE could preserve its intended incentive structure.
>
> However, this is not the setting considered in our paper. We model reward changes as external and potentially unknown to the agents, motivated by settings such as sensor drift, changing specifications, hardware degradation, or objective drift. In such cases, agents observe only their modified rewards and do not necessarily have access to a global reward-change function. Since MATE relies on a fixed token magnitude, preserving cooperation under reward-scale changes would require detecting the transformation and retuning or rescaling the token accordingly. DRIVE, in contrast, computes incentives directly from local reward differences, so its incentive magnitude changes automatically with the observed reward scale.
>
> We revised the manuscript to clarify this distinction. The intended comparison is not that MATE could never be made invariant under oracle parameter rescaling, but that fixed-token MATE is not invariant without additional global knowledge, synchronization, or meta-optimization. DRIVE provides invariance without such retuning because its exchange values are tied to the currently observed reward differences.
>
> ## Elementary affine lemma
>
> We also agree that the affine-preservation lemma is elementary. We retain it for completeness because it anchors the subsequent invariance proofs and makes the role of the positive affine assumption explicit. Its purpose is not to claim a surprising result, but to make the invariance argument self-contained.

---

### Author Response · Authors · 2026-05-27
**Rebuttal Summary**

We thank the reviewers for their constructive feedback, and we also thank the AE for coordinating the review process. We have uploaded a revised manuscript, with changes highlighted in blue, and provided individual responses to the reviews.

Briefly, the revision and rebuttals address the reviewer comments as follows:

* We clarified the intended scope of DRIVE as a mechanism for social dilemmas.
* We clarified the comparison to MATE: fixed-token MATE is not invariant without retuning, while DRIVE adapts through local reward differences.
* We expanded the limitations discussion on truthful communication, strategic misreporting, and protocol non-compliance.
* We added a continuous mean-field analysis and a finite-grid no-regret corollary to clarify the repeated-game convergence intuition, while more explicitly discussing how the existing graphical N-agent PD extension relates to the scope and limitations of the theory.
* We added stochastic reward-perturbation analysis and preliminary noisy-reward experiments.
* We added ablations on epoch-average reward estimation.
* We expanded the discussion of baseline diversity with preliminary MEDIATE results and commit to full MEDIATE and LToS comparisons in all main-paper scenarios in the final revision.
* We clarified the decentralized/ad hoc motivation, neighborhood effects, baseline hyperparameter choices, and several notation/formatting issues.

We appreciate the reviewers’ comments, which helped us improve the clarity, scope, and empirical support of the paper. We are happy to provide further clarifications if needed.

---

### Decision · Action_Editor_2wo4 · 2026-07-02

**Recommendation:** Accept with minor revision

**Additional Comments:**

Although the authors revised the paper, it is recommended that the authors further revise the paper for readability. For example, in Line 10 in Algorithm 1,  what is "Ext." ?  Same for "acc." in Algorithm 2.

**Audience:**

Yes

**Audience Explanation:**

Within the multi-agent (MA) community, there exist researchers who work on settings under social dilemma although many work on fully-cooperative MARL with fully shared rewards. Indeed, the MA problem under social dilemma captures many real-world problems in which each agent tries to maximize its own return, while too greedy policies may collapse the entire system or some agents are overly exploited. In this setting, the development of distributed mechanisms that enhance cooperation for system sustainability is an important issue, as evident by active research in peer incentivization. So, some researchers will interested in this work, although the proposed work seems simple and limited.

**Claims And Evidence:**

Yes

**Claims Explanation:**

In this paper,the authors proposed Dynamic Reward Incentives for Variable Exchange (DRIVE), a peer incentive mechanism, which exchanges the difference between the average reward and the reward at time t among local neighbors for multi-agent systems with social dilemma, and claimed that their algorithm maintains invariance under changing rewards modeled by affine transform and is robust agains per-agent reward perturbations.   The claims are supported by their theorems (Theorems 3, 4, 5 and 6) and experimental results.  The authors did not claim the covergence of the algorithm and clearly stated this as limitation.  So, it can be said that the claimed statements are well supported.